# Temporally-Extended $\epsilon$-Greedy Exploration

**Will Dabney, Georg Ostrovski & André Barreto**
DeepMind
London, UK
{wdabney,ostrovski,andrebarreto}@google.com

## Abstract

Recent work on exploration in reinforcement learning (RL) has led to a series of increasingly complex solutions to the problem. This increase in complexity often comes at the expense of generality. Recent empirical studies suggest that, when applied to a broader set of domains, some sophisticated exploration methods are outperformed by simpler counterparts, such as $\epsilon$-greedy. In this paper we propose an exploration algorithm that retains the simplicity of $\epsilon$-greedy while reducing dithering. We build on a simple hypothesis: the main limitation of $\epsilon$-greedy exploration is its lack of temporal persistence, which limits its ability to escape local optima. We propose a temporally extended form of $\epsilon$-greedy that simply repeats the sampled action for a random duration. It turns out that, for many duration distributions, this suffices to improve exploration on a large set of domains. Interestingly, a class of distributions inspired by ecological models of animal foraging behaviour yields particularly strong performance.

## 1 Introduction

Exploration is widely regarded as one of the most important open problems in reinforcement learning (RL). The problem has been theoretically analyzed under simplifying assumptions, providing reassurance and motivating the development of algorithms (Brafman and Tennenholtz, 2002; Asmuth et al., 2009; Azar, Osband, and Munos, 2017). Recently, there has been considerable progress on the empirical side as well, with new methods that work in combination with powerful function approximators to perform well on challenging large-scale exploration problems (Bellemare et al., 2016; Ostrovski et al., 2017; Burda et al., 2018; Badia et al., 2020b).

Despite all of the above, the most commonly used exploration strategies are still simple methods like $\epsilon$-greedy, Boltzmann exploration and entropy regularization (Peters, Mulling, and Altun, 2010; Sutton and Barto, 2018). This is true for both work of a more investigative nature (Mnih et al., 2015) and practical applications (Levine et al., 2016; Li et al., 2019). In particular, many recent successes of deep RL, from data-center cooling to Atari game playing, rely heavily upon these simple exploration strategies (Mnih et al., 2015; Lazic et al., 2018; Kapturowski et al., 2019).

Why does the RL community continue to rely on such naive exploration methods? There are several possible reasons. First, principled methods usually do not scale well. Second, the exploration problem is often formulated as a separate problem whose solution itself involves quite challenging steps. Moreover, besides having very limited theoretical grounding, practical methods are often complex and have significantly poorer performance outside a small set of domains they were specifically designed for. This last point is essential, as an effective exploration method must be generally applicable.

Naive exploration methods like $\epsilon$-greedy, Boltzmann exploration and entropy regularization are general because they do not make strong assumptions about the underlying domain. As a consequence, they are also *simple*, not requiring too much implementation effort or per-domain tuning. This makes them appealing alternatives even when they are not as efficient as some more complex variants.

Perhaps there is a middle ground between simple yet inefficient exploration strategies and more complex, though efficient, methods. The method we propose in this paper represents such a compromise. We ask the following question: how can we deviate minimally from the simple exploration strategies adopted in practice and still get clear benefits? In more pragmatic terms, we want a

simple-to-implement algorithm that can be used in place of naive methods and lead to improved exploration.

In order to achieve our goal we propose a method that can be seen as a generalization of $\epsilon$-greedy, perhaps the simplest and most widely adopted exploration strategy. As is well known, the $\epsilon$-greedy algorithm selects an exploratory action uniformly at random with probability $\epsilon$ at each time step. Besides its simplicity, $\epsilon$-greedy exploration has two properties that contribute to its universality:

1 It is *stationary*, i.e. its mechanics do not depend on learning progress. Stationarity is important for stability, since an exploration strategy interacting with the agent's learning dynamics results in circular dependencies that can in turn limit exploration progress. In simple terms: bad exploratory decisions can hurt the learned policy which can lead to more bad exploration.

2 It provides full *coverage* of the space of possible trajectories. All sequences of states, actions and rewards are possible under $\epsilon$-greedy exploration, albeit some with exceedingly small probability. This guarantees, at least in principle, that no solutions are excluded from consideration. Convergence results for RL algorithms rely on this sort of guarantee (Singh et al., 2000). This may also explain sophisticated exploration methods' use of $\epsilon$-greedy exploration (Bellemare et al., 2016).

However, $\epsilon$-greedy in its original form also comes with drawbacks. Since it does not explore persistently, the likelihood of deviating more than a few steps off the default trajectory is vanishingly small. This can be thought of as an inductive bias (or "prior") that favors transitions that are likely under the policy being learned (it might be instructive to think of a neighbourhood around the associated stationary distribution). Although this is not necessarily bad, it is not difficult to think of situations in which such an inductive bias may hinder learning. For example, it may be very difficult to move away from a local maximum if doing so requires large deviations from the current policy.

The issue above arises in part because $\epsilon$-greedy provides little flexibility to adjust the algorithm's inductive bias to the peculiarities of a given problem. By tuning the algorithm's only parameter, $\epsilon$, one can make deviations more or less likely, but the *nature* of such deviations is not modifiable. To see this, note that all sequences of exploratory actions are equally likely under $\epsilon$-greedy, regardless of the specific value used for $\epsilon$. This leads to a coverage of the state space that is largely defined by the current ("greedy") policy and the environment dynamics (see Figure 1 for an illustration).

In this paper we present an algorithm that retains the beneficial properties of $\epsilon$-greedy while at the same time allowing for more control over the nature of the induced exploratory behavior. In order to achieve this, we propose a small modification to $\epsilon$-greedy: we replace actions with temporally-extended sequences of actions, or *options* (Sutton, Precup, and Singh, 1999). Options then become a mechanism to modulate the inductive bias associated with $\epsilon$-greedy. We discuss how by appropriately defining a set of options one can "align" the exploratory behavior of $\epsilon$-greedy with a given environment or class of environments; we then show how a very simple set of domain-agnostic options work surprisingly well across a variety of well known environments.

## 2   BACKGROUND AND NOTATION

Reinforcement learning can be set within the Markov Decision Process (MDP) formalism (Puterman, 1994). An MDP $\mathcal{M}$ is defined by the tuple $(\mathcal{X}, \mathcal{A}, P, R, \gamma)$, where $x \in \mathcal{X}$ is a state in the state space, $a \in \mathcal{A}$ is an action in the action space, $P(x' \mid x, a)$ is the probability of transitioning from state $x$ to state $x'$ after taking action $a$, $R: \mathcal{X} \times \mathcal{A} \to \mathbb{R}$ is the reward function and $\gamma \in [0, 1)$ is the discount factor. Let $\mathscr{P}(\mathcal{A})$ denote the space of probability distributions over actions; then, a policy $\pi: \mathcal{X} \to \mathscr{P}(\mathcal{A})$ assigns some probability to each action conditioned on a given state. We will denote by $\pi_a = \mathbb{1}_a$ the policy which takes action $a$ deterministically in every state.

The agent attempts to learn a policy $\pi$ that maximizes the expected return or value in a given state,

$$V^\pi(x) = \mathbb{E}_{A \sim \pi} Q^\pi(x, A) = \mathbb{E}_\pi \left[ \sum_{t=0}^\infty \gamma^t R(X_t, A_t) \mid X_0 = x \right],$$

where $V^\pi$, $Q^\pi$ are the value and action-value functions of $\pi$. The *greedy* policy for action-value function $Q$ takes the action $\arg \max_{a \in \mathcal{A}} Q(x, a), \forall x \in \mathcal{X}$. In this work we primarily rely upon methods based on the $Q$-learning algorithm (Watkins and Dayan, 1992), which attempts to learn the

(a) $\epsilon$–Greedy    (b) Temporally–extended $\epsilon$–Greedy

$\epsilon = 0.1$    $\epsilon = 0.5$    $\epsilon = 0.9$    $\epsilon = 1.0$    $\epsilon = 0.1$    $\epsilon = 0.5$    $\epsilon = 0.9$    $\epsilon = 1.0$

Figure 1: Average (estimated) first-visit times, comparing $\epsilon$-greedy policies **(a)** without and with **(b)** temporal persistence, in an open gridworld (blue represents fewer steps to and red states rarely or never seen). Greedy policy moves directly down from the top center. See Appendix for details.

optimal policy by approximating the Bellman optimality equation:

$$Q(x,a) = R(x,a) + \gamma \mathbb{E}_{X' \sim P(\cdot|x,a)} \left[ \max_{a' \in \mathcal{A}} Q(X', a') \right]. \tag{1}$$

In practice, the state space $\mathcal{X}$ is often too large to represent exactly and thus we have $Q_\theta(x,a) \approx Q(x,a)$ for a function approximator parameterized by $\theta$. We will generally use some form of differentiable function approximator $Q_\theta$, whether it be linear in a fixed set of basis functions, or an artificial neural network. We update parameters $\theta$ to minimize a squared or Huber loss between the left- and right-hand sides of equation 1, with the right-hand side held fixed (Mnih et al., 2015).

In addition to function approximation, it has been argued that in order to scale to large problems, RL agents should be able to reason at multiple temporal scales (Dayan and Hinton, 1993; Parr and Russell, 1998; Sutton, Precup, and Singh, 1999; Dietterich, 2000). One way to model temporal abstraction is via *options* (Sutton, Precup, and Singh, 1999), i.e. temporally-extended courses of action. In the most general formulation, an option can depend on the entire *history* between its initiation time step $t$ and the current time step $t + k$, $h_{t:t+k} \equiv x_t a_t x_{t+1}...a_{t+k-1}x_{t+k}$. Let $\mathcal{H}$ be the space of all possible histories; a *semi-Markov option* is a tuple $\omega \equiv (\mathcal{I}_\omega, \pi_\omega, \beta_\omega)$, where $\mathcal{I}_\omega \subseteq \mathcal{X}$ is the set of states where the option can be initiated, $\pi_\omega \colon \mathcal{H} \to \mathscr{P}(\mathcal{A})$ is a history-dependent policy, and $\beta_\omega \colon \mathcal{H} \mapsto [0,1]$ gives the probability that the option terminates after observing some history (Sutton, Precup, and Singh, 1999). As in this work we will use options for exploration, we will assume that $\mathcal{I}_\omega = \mathcal{X}, \forall \omega$.

Once an option $\omega$ is selected, the agent takes actions $a \sim \pi_\omega(\cdot \mid h)$ after having observed history $h \in \mathcal{H}$ and at each step terminates the option with probability $\beta_\omega(h)$. It is worth emphasizing that semi-Markov options depend on the history since their initiation, but not before. Also, they are usually defined with respect to a statistic of histories $h \in \mathcal{H}$; for example, by looking at the *length* of $h$ one can define an option that terminates after a fixed number of steps.

## 3  EXPLORATION IN REINFORCEMENT LEARNING

At its core, RL presents the twin challenges of temporal credit assignment and exploration. The agent must accurately, and efficiently, assign credit to past actions for their role in achieving some long-term return. However, to continue improving the policy, it must also consider behaviours it estimates to be sub-optimal. This leads to the well-known *exploration-exploitation trade-off*.

Because of its central importance in RL, exploration has been among the most studied topics in the field. In finite state-action spaces, the theoretical limitations of exploration, with respect to sample complexity bounds, are fairly well understood (Azar, Osband, and Munos, 2017; Dann, Lattimore, and Brunskill, 2017). However, these results are of limited practical use for two reasons. First, they bound sample complexity by the size of the state-action space and horizon, which makes their immediate application in large-scale or continuous state problems difficult. Second, these algorithms tend to be designed based on worst-case scenarios, and can be inefficient on problems of actual interest. Bayesian RL methods for exploration address the explore-exploit problem integrated with the estimation of the value-function itself (Kolter and Ng, 2009). Generally such methods strongly depend upon the quality of their priors, which can be difficult to set appropriately. Thompson sampling methods (Thompson, 1933; Osband, Russo, and Van Roy, 2013) estimate the posterior distribution of value-functions, and act greedily according to a sample from this distribution. As with other methods which integrate learning and exploration into a single estimation problem, this creates non-stationary, but temporally persistent, exploration. Other examples of this type of exploration strategy include randomized prior functions (Osband, Aslanides, and Cassirer, 2018), uncertainty Bellman equations (O'Donoghue et al., 2018), NoisyNets (Fortunato et al., 2017), and successor uncertainties (Janz

et al., 2019). Although quite different from each other, they share key commonalities: non-stationary targets, temporal persistence, and exploration based on the space of value functions.

At the other end of the spectrum, there have recently been successful attempts to design algorithms with specific problems of interest in mind. Certain games from the Atari-57 benchmark (e.g. MONTEZUMA'S REVENGE, PITFALL!, PRIVATE EYE) have been identified as 'hard exploration games' (Bellemare et al., 2016), attracting the attention of the research community, leading to significant progress in terms of performance (Ecoffet et al., 2019; Burda et al., 2018). On the downside, these results have been usually achieved by algorithms with little or no theoretical grounding, adopting specialized inductive biases, such as density modeling of images (Bellemare et al., 2016; Ostrovski et al., 2017), error-seeking intrinsic rewards (Pathak et al., 2017; Badia et al., 2020a), or perfect deterministic forward-models (Ecoffet et al., 2019).

Generally, such algorithms are evaluated only on the very domains they are designed to perform well on, raising questions of generality. Recent empirical analysis showed that some of these methods perform similarly to each other on hard exploration problems and significantly under-perform $\epsilon$-greedy otherwise (Ali Taïga et al., 2020). One explanation is that complex algorithms tend to be more brittle and harder to reproduce, leading to lower than expected performance in follow-on work. However, these results also suggest that much of the recent work on exploration is over-fitting to a small number of domains.

## 4 TEMPORALLY-EXTENDED EXPLORATION

There are many ways to think about exploration: curiosity, experimentation, reducing uncertainty, etc. Consider viewing exploration as a search for undiscovered rewards or shorter paths to known rewards. In this context, the behaviour of $\epsilon$-greedy appears shortsighted because the probability of moving consistently in any direction decays exponentially with the number of exploratory steps. In Figure 1a we visualize the behaviour of uniform $\epsilon$-greedy in an open gridworld, where the agent starts at the center-top and the greedy policy moves straight down. Observe that for $\epsilon \leq 0.5$ the agent is exceedingly unlikely to reach states outside a narrow band around the greedy policy. Even the purely exploratory policy ($\epsilon = 1.0$) requires a large number of steps to visit the bottom corners of the grid. This is because, under the uniform policy, the probability of moving consistently in any direction decays exponentially (see Figure 1a). By contrast, a method that explores persistently with a directed policy leads to more efficient exploration of the space at various values of $\epsilon$ (Figure 1b).

The importance of temporally-extended exploration has been previously highlighted (Osband et al., 2016), and in general, count-based (Bellemare et al., 2016) or curiosity-based (Burda et al., 2018) exploration methods are inherently temporally-extended due to integrating exploration and exploitation into the greedy policy. Here our goal is to leverage the benefits of temporally-extended exploration without modifying the greedy policy.

There has been a wealth of research on learning options (McGovern and Barto, 2001; Stolle and Precup, 2002; Şimşek and Barto, 2004; Bacon, Harb, and Precup, 2017; Harutyunyan et al., 2019), and specifically for exploration (Machado, Bellemare, and Bowling, 2017; Machado et al., 2018b; Jinnai et al., 2019; 2020; Hansen et al., 2020). These methods use options for exploration and to augment the action-space, adding learned options to actions available at states where they can be initiated.

In the remainder of this work, we argue for temporally-extended exploration, using options to encode a set of inductive biases to improve sample-efficiency. This fundamental message is found throughout the existing work on exploration with options, but producing algorithms that are empirically effective on large environments remains a challenge for the field. In the next section, we discuss in more detail how the options' policy $\pi_\omega$ and termination $\beta_\omega$ can be used to induce different types of exploration.

**Temporally-Extended $\epsilon$-Greedy** A *temporally-extended $\epsilon$-greedy* exploration strategy depends on choosing an exploration probability $\epsilon$, a set of options $\Omega$, and a sampling distribution $p$ with support $\Omega$. On each step the agent follows the current policy $\pi$ for one step with probability $1 - \epsilon$, or with probability $\epsilon$ samples an option $\omega \sim p(\Omega)$ and follows it until termination. Standard $\epsilon$-greedy has three desirable properties that help explain its wide adoption in practice: it is *simple*, *stationary*, and promotes full *coverage* of the state-action space in the limit (guaranteeing convergence to the optimal policy under the right conditions). We now discuss to what extent the proposed algorithm retains

these properties. Although somewhat subjective, it seems fair to call temporally-extended $\epsilon$-greedy a simple method. It is also stationary when the set of options $\Omega$ and distribution $p$ are fixed, for in this case its mechanics are not influenced by the collected data. Finally, it is easy to define conditions under which temporally-extended $\epsilon$-greedy covers the entire state-action space, as we discuss next.

Obviously, the exploratory behavior of temporally-extended $\epsilon$-greedy will depend on the set of options $\Omega$. Ideally we want all actions $a \in \mathcal{A}$ to have a nonzero probability of being executed in all states $x \in \mathcal{X}$ regardless of the greedy policy $\pi$. This is clearly not the case for all sets $\Omega$. In fact, this may not be the case even if for all $(x, a) \in \mathcal{X} \times \mathcal{A}$ there is an option $\omega \in \Omega$ such that $\pi_\omega(a|hx) > 0$, where $hx$ represents all histories ending in $x$. To see why, note that, given a fixed $\Omega$ and $\epsilon > 0$, it may be impossible for an option $\omega \in \Omega$ to be "active" in state $x$ (that is, either start at or visit $x$). For example, if all options in $\Omega$ terminate after a fixed number of steps that is a multiple of $k$, temporally-extended $\epsilon$-greedy with $\epsilon = 1$ will only visit states of an unidirectional chain whose indices are also multiples of $k$. Perhaps even subtler is that, even if all options can be active at state $x$, the *histories* $hx \in \mathcal{H}$ associated with a given action $a$ may themselves not be realizable under the combination of $\Omega$ and the current $\pi$.

It is clear then that the coverage ability of temporally-extended $\epsilon$-greedy depends on the interaction between $\pi$, $\Omega$, $\epsilon$, and the dynamics $P(\cdot|x, a)$ of the MDP. One way to reason about this is to consider that, once fixed, these elements induce a stochastic process which in turn gives rise to a well-defined distribution over the space of histories $\mathcal{H}$.

**Property 1** (Full coverage). Let $\mathcal{M}$ be the space of all MDPs with common state-action spaces $\mathcal{X}$, $\mathcal{A}$, and $\Omega$ a set of options defined over this state-action space. Then, $\Omega$ has **full coverage** for $\mathcal{M}$ if $\forall M \in \mathcal{M}$, $\epsilon > 0$, and $\pi$, the semi-Markov policy $\mu := (1 - \epsilon)\pi + \epsilon\pi_\omega$, where $\omega$ is a random variable uniform over $\Omega$, visits every state-action pair with non-zero probability. Note that $\mu$ is itself a random variable and not an average policy.

We can then look for simple conditions that would lead to having Property 1. For example, if the options' policies only depend on the last state of the history, $\pi_\omega(\cdot|hx) = \pi_\omega(\cdot|x)$ (i.e. they are Markov, rather than semi-Markov policies), we can get the desired coverage by having $\pi_\omega(a|x) > 0$ for all $x \in \mathcal{X}$ and all $a \in \mathcal{A}$. The coverage of $\mathcal{X} \times \mathcal{A}$ also trivially follows from having all primitive actions $a \in \mathcal{A}$ as part of $\Omega$. Note that if the primitive actions are the *only* elements of $\Omega$ we recover standard $\epsilon$-greedy, and thus coverage of $\mathcal{X} \times \mathcal{A}$. Of course, in these and similar cases, temporally-extended $\epsilon$-greedy allows for convergence to the optimal policy under the same conditions as its precursor.

This view of temporally-extended $\epsilon$-greedy, as inducing a stochastic process, also helps us to understand its differences with respect to its standard counterpart. Since the induced stochastic process defines a distribution over histories we can also talk about distributions over sequences of actions. With standard $\epsilon$-greedy, every sequence of $k$ exploratory actions has a probability of occurrence of exactly $(\epsilon/|\mathcal{A}|)^k$, where $|\mathcal{A}|$ is the size of the action space. By changing $\epsilon$ one can uniformly change the probabilities of *all* length-$k$ sequences of actions, but no sequence can be favored over the others. Temporally-extended $\epsilon$-greedy provides this flexibility through the definition of $\Omega$; specifically, by defining the appropriate set of options one can control the temporal correlation between actions. This makes it possible to control how *quickly* the algorithm converges, as we discuss next.

**Efficient exploration**     For sample-efficiency we want to cover the state-action space *quickly*.

**Definition 1.** The **cover time** of an RL algorithm is the number of steps needed to visit all state-action pairs at least once with probability $0.5$ starting from the initial state distribution.

Even-Dar and Mansour (2003) show that the sample efficiency of Q-learning can be bounded in terms of the cover time of the exploratory policy used. Liu and Brunskill (2018) provide an upper bound for the cover time of a random exploratory policy based on properties of the MDP. Putting these results together, we have the characterization of a class of MDPs for which Q-learning plus $\epsilon$-greedy exploration is sample efficient (that is, it converges in polynomial time).

Normally, the efficiency of $\epsilon$-greedy Q-learning is completely determined by the MDP: given an MDP, either the algorithm is efficient or it is not. We now discuss how by replacing $\epsilon$-greedy exploration with its temporally-extended counterpart we can have efficient exploration on a much broader class of MDPs. To understand why this is so, note that the definition of the set of options $\Omega$ can be seen as the definition of a new MDP in which histories play the role of states and options play the role of actions.

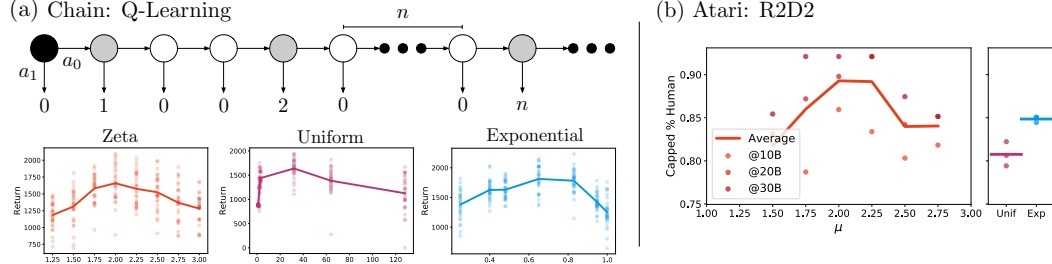

Figure 2: **(a)** Modified chain MDP, action $a_0$ moves right, $a_1$ terminates with specified reward. Rewards follow a pattern of n zeros followed by a single reward $n$, etc. Evaluation of performance under various duration distributions and hyper-parameters on the chain. **(b)** Duration distribution similarly compared for an R2D2-based deep RL agent in Atari.

Hence, by appropriately defining $\Omega$, we can have an MDP in which random exploration is efficient. We now formalize this notion by making explicit properties of $\Omega$ that lead to efficient exploration:

**Assumption 1.** For an MDP $M$ and set of options $\Omega$, there exists $n_{\max} \in \mathbb{N}$ such that $\forall x, y \in \mathcal{X}$, $\exists \omega \in \Omega$ leading to $\mathbb{E}_{\pi_\omega}[t \mid x_0 = x, x_t = y] \leq n_{\max}$.

**Theorem 1.** For any irreducible MDP, let $\Omega$ be a set of options satisfying Assumption 1 with $n_{max} \leq \Theta(|\mathcal{X}||\mathcal{A}|)$. Then, temporally-extended $\epsilon$-greedy with sampling distribution $p$ satisfying $\frac{1}{\rho(\omega)} \leq \Theta(|\mathcal{X}||\mathcal{A}|), \forall \omega \in \Omega$, has polynomial sample complexity.

In many cases it is easy to define options that satisfy Assumption 1, as we will discuss shortly. But even when this is not the case, one can *learn* options deliberately designed to have this property. For example, Jinnai et al. (2019; 2020) learn point-options (transitioning from one state to one other state) that explicitly minimize cover time. The approach proposed by Machado et al. (2017; 2018b) also leads to options with a small cover time. Alternatively, Whitney et al. (2020) learn an embedding of action sequences such that sequences with similar representations also have similar future state distributions. They observe that sampling uniformly in this abstracted action space yields action sequences whose future state distribution is nearly uniform over reachable states. We can interpret such an embedding, coupled with a corresponding decoder back into primitive actions, as an appealing approach to learning open-loop options for temporally-extended $\epsilon$-greedy exploration.

Next, we propose a concrete form of temporally-extended $\epsilon$-greedy which requires neither learning $\Omega$ nor specific domain knowledge. These options encode a commonly held inductive bias: actions have (largely) consistent effects throughout the state-space.

$\epsilon z$**-Greedy** We begin with the options $\omega_a \equiv (\mathcal{X}, \pi_a, \beta)$, where $\pi_a(h) = \mathbb{1}_a$ and $\beta(h) = 1$ for all $h \in \mathcal{H}$, and consider a single modification, temporal persistence. Let $\omega_{an} \equiv (\mathcal{X}, \pi_a, \beta(h) = \mathbb{1}_{|h|==n})$ be the option which takes action $a$ for $n$ steps and then terminates. Our proposed algorithm, is to let $\Omega = \{\omega_{an}\}_{a \in \mathcal{A}, n \geq 1}$ and $p$ to be uniform over actions with durations distributed according to some distribution $z$. Intuitively, we are proposing the set of semi-Markov options made up of all "action-repeat" policies for all combinations of actions and repeat durations, with a parametric sampling distribution on durations.

This exploration algorithm is described by two parameters, $\epsilon$ dictating when to explore, and $z$ dictating the degree of persistence. Notice that when $z$ puts all mass on $n = 1$, this is standard $\epsilon$-greedy; more generally this combination of distributions forms a composite distribution with support $[0, \infty)$, which is to say that with some probability the agent explores for $n = 0$ steps, corresponding to following its usual policy, and for all other $n > 0$ the agent explores following an action-repeat policy.

A natural question arises: what distribution over durations should we use? To help motivate this question, and to help understand the desirable characteristics, consider Figure 2 which shows a modified chain MDP with two actions. Taking the 'down' action immediately terminates with the specified reward, whereas taking the 'right' action progresses to the next state in the chain. Similar to other exploration chain-like MDPs, $\epsilon$-greedy performs poorly here because the agent must move consistently in one direction for an arbitrary number of steps (determined by the discount) to reach the optimal reward. Instead, we consider the effects of three classes of duration distribution: exponential

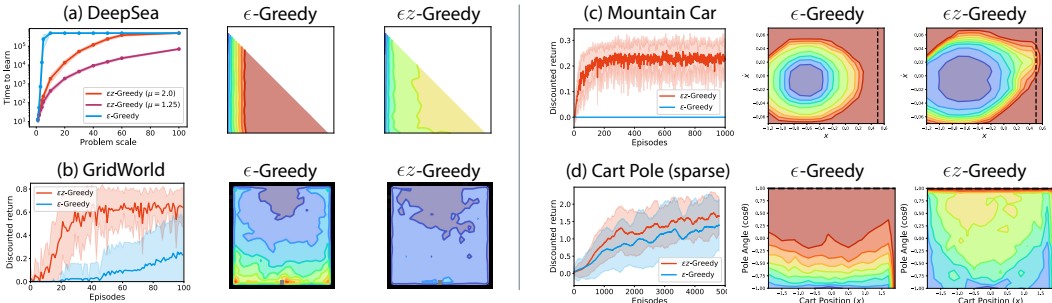

Figure 3: Comparing $\epsilon$-greedy with $\epsilon z$-greedy on four small-scale domains requiring exploration. (a) DeepSea is a tabular problem in which only one action-sequence receives positive reward, and uniform exploration is exponentially inefficient, (b) GridWorld is a four-action gridworld with a single reward, (c) MountainCar is the sparse reward (only at goal) version of the classic RL domain, and (d) CartPole swingup-sparse only gives non-zero reward when the pole is perfectly balanced and the cart near-center. For each, we show performance comparing $\epsilon$-greedy with $\epsilon z$-greedy (left), as well as average first-visit times over states for both algorithms during **pure exploration** ($\epsilon = 1$). In all first-visit plots, color levels are linearly scaled, except for DeepSea in which we use a log scale.

($z(n) \propto \lambda^{n-1}$), uniform ($z(n) = \mathbb{1}_{n \leq N}/N$), and zeta ($z(n) \propto n^{-\mu}$). Figure 2b shows the average return achieved by these distributions as their hyper-parameters are varied. This problem illustrates that, without prior knowledge of the MDP, it is important to support long durations, such as with a heavy-tailed distribution (e.g. the zeta distribution).

Why not simply allow uniform over an extremely large support? Doing so would effectively force 'pure' exploration without any exploitation, because this form of *ballistic* exploration would simply continue exploring indefinitely. Indeed, we can see in Figure 2 that the uniform distribution leads to poor performance (the same is true for the zeta distribution as $\mu \to 1$, which also leads to ballistic exploration). On the other hand, short durations lead to frequent switching and vanishingly small probabilities of reaching larger rewards at all. This trade-off leads to the existence of an optimal value of $\mu$ for the zeta distribution that can vary by domain (Humphries et al., 2010), and is illustrated by the inverted U-curve in Figure 2. A class of ecological models for animal foraging known as *Lévy flights* follow a similar pattern of choosing a direction uniformly at random, and following that direction for a duration sampled from a heavy-tailed distribution. Under certain conditions, this has been shown to be an optimal foraging strategy, a form of exploration for a food source of unpredictable location (Viswanathan et al., 1996; 1999). In particular, a value of $\mu = 2$ has consistently been found as the best for modeling animal foraging, as well as performing best in our hyper-parameter sweep. Thus, in the remainder of this work we will use the zeta distribution, with $\mu = 2$ unless otherwise specified, and call this combination of $\epsilon$ chance to explore and zeta-distributed durations, $\epsilon z$-Greedy exploration[1].

## 5 EXPERIMENTAL RESULTS

We have emphasized the importance of simplicity, generality (via convergence guarantees), and stationarity of exploration strategies. We proposed a simple temporally-extended $\epsilon$-greedy algorithm, $\epsilon z$-greedy, and saw that a heavy-tailed duration distribution yielded the best trade-off between temporal persistence and sample efficiency. In this section, we present empirical results on tabular, linear, and deep RL settings, pursuing two objectives: The first is to demonstrate the generality of our method in applying it across domains as well as across multiple value-based reinforcement learning algorithms (Q-learning, SARSA, Rainbow, R2D2). Second, we make the point that exploration comes at a cost, and that $\epsilon z$-greedy improves exploration with significantly less loss in efficiency on dense-reward domains compared with existing exploration algorithms.

**Small-Scale: Tabular & Linear RL**   We consider four small-scale environments (DeepSea, Grid-World, MountainCar, CartPole swingup-sparse), configured to be challenging sparse-reward explo-

---

[1]Pronounce 'easy-greedy'.

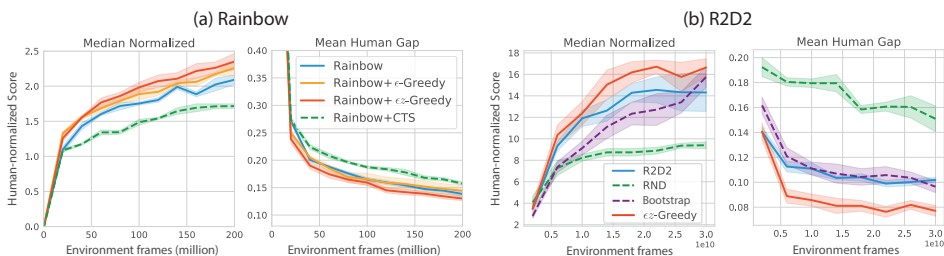

Figure 4: Results on the Atari-57 benchmark for (a) Rainbow-based and (b) R2D2-based agents.

ration problems. Full details for each are given in the Appendix. DeepSea and GridWorld use a tabular representation while MountainCar and CartPole use linear function approximation on an order 5 and 7 Fourier basis respectively (Konidaris, Osentoski, and Thomas, 2011).

In Figure 3 we present results comparing $\epsilon$-greedy and $\epsilon z$-greedy on these four domains. Unless otherwise specified, the hyper-parameters and training settings for these two methods are identical. For each domain we show (i) learning curves showing average return against training episodes[2], (ii) average first-visit times on states during pure ($\epsilon = 1.0$) exploration for $\epsilon$-greedy and (iii) $\epsilon z$-greedy. The results show that $\epsilon z$-greedy provides significantly improved performance on these domains, and the first-visit times indicate significantly better state-space coverage compared to $\epsilon$-greedy.

**Atari-57: Deep RL** Motivated by the results in tabular and linear settings, we now turn to deep RL and evaluate performance on 57 Atari 2600 games in the Arcade Learning Environment (ALE) (Bellemare et al., 2013). To demonstrate the generality of the approach, we apply $\epsilon z$-greedy to two state-of-the-art deep RL agents, Rainbow (Hessel et al., 2018) and R2D2 (Kapturowski et al., 2019). We compare with baseline performance as well as the performance of a recent intrinsic motivation-based exploration algorithm: CTS-based pseudo-counts (Bellemare et al., 2016) in Rainbow and RND (Burda et al., 2018) in R2D2, each tuned for performance comparable with published results. Finally, in R2D2 experiments we also compare with a Bootstrapped DQN version of R2D2 (Osband et al., 2016), providing an exploration baseline without intrinsic rewards. We include pseudo-code and hyper-parameters in the Appendix, though the implementation of $\epsilon z$-greedy in each case is trivial, hyper-parameters are mostly identical to previous work, and we fix $\mu = 2$ for results in this section. Our findings (see Figure 4) show that $\epsilon z$-greedy improves performance on the hard exploration tasks with little to no loss in performance on the rest of the suite. By comparison, we observe that the intrinsic motivation methods often (although not always) outperform $\epsilon z$-greedy on the hard exploration tasks, but at a significant loss of performance on the rest of the benchmark.

The results in Figure 4 show median human-normalized score over the 57 games and the human-gap, measuring how much the agent under-performs humans on average (see Appendix D for details). We consider the median to indicate overall performance on the suite and the human-gap to illustrate gains on the hard exploration games where agents still under-perform relative to humans, with full per-game and mean performance given in the Appendix. Table 1 gives the final performance of each of the agents in terms of these summary statistics. Figure 5 shows representative examples of per-game performance for the R2D2-based agents. These per-game results make a strong point, that even on the hard exploration games the inductive biases of intrinsic motivation methods may be poorly aligned, and that outside a small number of games these methods significantly hurt performance, whereas our proposed method improves exploration while avoiding this significant loss elsewhere.

To demonstrate that the effectiveness of our method does not crucially depend on evaluation on deterministic domains, in Appendix E we additionally show a similar comparison of the Rainbow-based agents on a stochastic variant of the Atari-57 benchmark using 'sticky actions' (Machado et al., 2018a). The results are qualitatively similar: while all agent variants do somewhat worse on the stochastic compared to the deterministic case, $\epsilon z$-greedy improves over the baseline and $\epsilon$-greedy on the hardest exploration domains while not substantially affecting performance in others, and coming out on top in terms of mean and median human-normalized performance as well as human-gap (Table 5, Figures 9, 10, 21).

---

[2]To compare with previous work on DeepSea, we report expected time to learn versus problem scale.

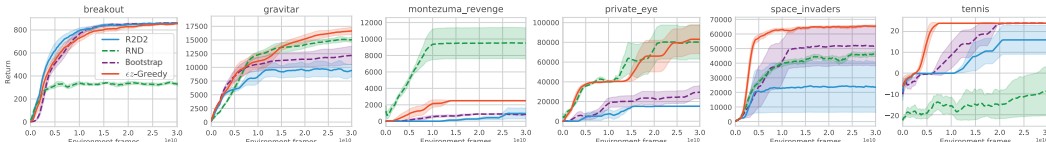

Figure 5: Results on the Atari-57 selected games showing R2D2-based agents.

| Algorithm (@30B) | Median | Mean | H-Gap | Algorithm (@200M) | Median | Mean | H-Gap |
|---|---|---|---|---|---|---|---|
| R2D2 | 14.31 | 39.55 | 0.102 | Rainbow | 2.09 | 8.82 | 0.139 |
| R2D2+RND | 9.40 | **42.17** | 0.151 | Rainbow+$\epsilon$-Greedy | 2.26 | 9.17 | 0.143 |
| R2D2+Bootstrap | 15.75 | 37.69 | 0.096 | Rainbow+CTS | 1.72 | 6.77 | 0.150 |
| R2D2+$\epsilon z$-greedy | **16.64** | 40.16 | **0.077** | Rainbow+$\epsilon z$-Greedy | **2.35** | **9.34** | **0.130** |

Table 1: Atari-57 final performance. H-Gap denotes the human-gap, defined fully in the Appendix.

## 6 DISCUSSION AND CONCLUSIONS

We have proposed temporally-extended $\epsilon$-greedy, a form of random exploration performed by sampling an option and following it until termination, with a simple instantiation which we call $\epsilon z$-greedy. We showed, across domains and algorithms spanning tabular, linear and deep reinforcement learning that $\epsilon z$-greedy improves exploration and performance in sparse-reward environments with minimal loss in performance on easier, dense-reward environments. Further, we showed that compared with other exploration methods (pseudo-counts, RND, Bootstrap), $\epsilon z$-greedy has comparable performance averaged over the hard-exploration games in Atari, but without the significant loss in performance on the remaining games. Although action-repeats have been a part of deep RL algorithms since DQN, and have been considered as a type of option (Schoknecht and Riedmiller, 2002; 2003; Braylan et al., 2015; Lakshminarayanan, Sharma, and Ravindran, 2017; Sharma, Lakshminarayanan, and Ravindran, 2017), their use for exploration with sampled durations does not appear to have been studied before.

**Generality and Limitations.** Both $\epsilon$- and $\epsilon z$-greedy are guaranteed to converge in the finite state-action case, but they place probability mass over exploratory trajectories very differently, thus encoding different inductive biases. We expect there to be environments where $\epsilon z$-greedy significantly under-performs $\epsilon$-greedy. Indeed, these are easy to imagine: DeepSea with action effects randomized per-state (see Appendix Figure 14), GridWorld with many obstacles that immediately end the episode ('mines'), a maze changing direction every few steps, etc. More generally, the limitations of $\epsilon z$-greedy are: **(i)** Actions may not homogeneously (over states) correspond to a natural notion of shortest-path directions in the MDP. **(ii)** Action spaces may be biased (e.g. many actions have the same effect), so uniform action sampling may produce undesirable biased drift through the MDP. **(iii)** Obstacles and dynamics in the MDP can cause long exploratory trajectories to waste time (e.g. running into a wall for thousands of steps), or produce uninformative transitions (e.g. end of episode, death). In Appendix F we report on a series of experiments investigating $\epsilon z$-Greedy's sensitivities to such modifications of the Gridworld domain and find that its performance degrades gracefully overall.

These limitations are precisely where we believe future work is best motivated. How can an agent learn stationary, problem-specific notions of direction, and explore in that space efficiently? How to avoid wasteful long trajectories, perhaps by truncating early? This form of exploration bears similarity to the Lévy-flights model of foraging, where an animal will abruptly end foraging as soon as food is within sight. Could we use discrepancies in value along a trajectory to similarly truncate exploration early? Recent work around learning action representations appear to be promising directions (Tennenholtz and Mannor, 2019; Chandak et al., 2019).

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

# APPENDICES

## A Cover Time Analysis

**Assumption 1.** For an MDP $M$ and set of options $\Omega$, there exists $n_{\max} \in \mathbb{N}$ such that $\forall x, y \in \mathcal{X}$, $\exists \omega \in \Omega$ leading to $\mathbb{E}_{\pi_\omega}[t \mid x_0 = x, x_t = y] \leq n_{\max}$.

**Theorem 1.** For any irreducible MDP, let $\Omega$ be a set of options satisfying Assumption 1 with $n_{max} \leq \Theta(|\mathcal{X}||\mathcal{A}|)$. Then, temporally-extended $\epsilon$-greedy with sampling distribution $p$ satisfying $\frac{1}{\rho(\omega)} \leq \Theta(|\mathcal{X}||\mathcal{A}|), \forall \omega \in \Omega$, has polynomial sample complexity.

*Proof.* Liu and Brunskill (2018) establish a PAC RL bound, leading to polynomial sample complexity, for random exploration using primitive actions when $\frac{1}{\min_x \phi(x)}$ and $\frac{1}{h}$ are polynomial in states and actions, with steady-state distribution $\phi$ and Cheeger constant $h$. However, their result does not require actions to be primitive, and here we show how temporally-extended actions in the form of exploratory options can be used for a similar result.

We begin by bounding the steady-state probability for any state $x \in \mathcal{X}$, where $x_0 := \arg\max_x \phi(x)$ and $n_{\max}$ the maximum expected path distance between two states. We can understand this as bounding the steady-state probability of $x$ by the product of (1) the maximal steady-state probability over states, (2) the probability of choosing an option from $x_0$ that reaches $x$. Let $\omega$ be any option satisfying Assumption 1 for starting in state $x_0$ and reaching $x$,

$$\phi(x) \geq \phi(x_0) \times p(\omega), \quad \implies \quad \frac{1}{\min_x \phi(x)} \leq \frac{\Theta(|\mathcal{X}||\mathcal{A}|)}{\phi(x_0)}.$$

Next, we can similarly bound the Cheeger constant. Recall from (Liu and Brunskill, 2018) that $h = \inf_U \frac{F(\partial U)}{\min\{F(U), F(\bar{U})\}}$ where $\bar{U}$ denotes the set of states not in $U$,

$$F(u, v) = \phi(u)P(u, v), \quad F(\partial U) = \sum_{u \in U, v \notin U} F(u, v), \quad F(U) = \sum_{u \in U} \phi(u).$$

Let $U' = \{x_0\}$, then, the Cheeger constant can be bounded by,

$$
\begin{aligned}
h &= \inf_U \frac{F(\partial U)}{\min\{F(U), F(\bar{U})\}}, \\
&\geq \frac{F(\partial U')}{\min\{F(U'), F(\bar{U'})\}}, \\
&\geq \frac{\sum_{x \neq x_0} \phi(x_0) P(x_0, x)}{\phi(x_0)}, \\
&= \sum_{x \neq x_0} P(x_0, x), \\
&\geq p(\omega), \quad \implies \\
\frac{1}{h} &\leq \Theta(|\mathcal{X}||\mathcal{A}|).
\end{aligned}
$$

$\square$

### Example: Chain MDP

Theorem 1 clarifies the conditions under which temporally-extended $\epsilon$-greedy is efficient. Given an MDP, this will depend on two factors: the options and the sampling distribution. To illustrate this point, we use the well known Chain MDP, for which $\epsilon z$-Greedy satisfies Assumption 1. Specifically, the requirement on $z$ is satisfied by the zeta distribution ($z(n) = \frac{n^{-\mu}}{\zeta(\mu)}$) but not by the geometric distribution (exponential decay). This implies that $\epsilon z$-greedy on the Chain MDP will have polynomial sample complexity when $z$ is zeta distributed, but not when exponentially distributed.

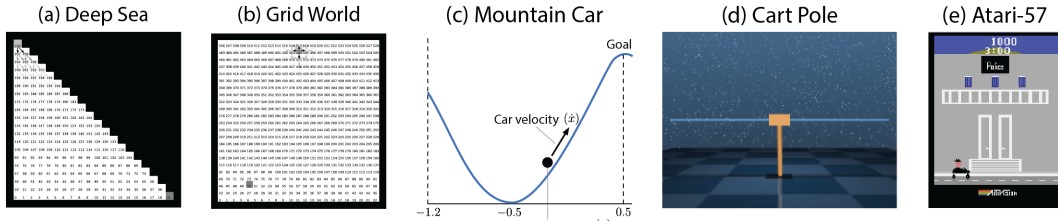

Figure 6: Environments used in this work: **(a)** DeepSea, **(b)** GridWorld, **(c)** MountainCar, **(d)** CartPole, **(e)** Atari-57.

We can see this by observing that in a Chain MDP of size $N$ any state can be reached from the starting state within at most $N$ steps, yielding $n_{max} \leq |\mathcal{X}|$. The sampling distribution $\rho$ is uniform over actions meaning that $\rho(\omega) \geq z(n_{max})/|\mathcal{A}|$. Finally, we consider the specific form of the duration distribution $z$. When given by a zeta distribution, $z(n) = n^{-\mu}/\zeta(\mu)$, we have

$$
\begin{aligned}
\frac{1}{\rho(\omega)} &\leq \frac{|\mathcal{A}|}{z(n_{max})}, \\
&= \frac{|\mathcal{A}|\zeta(\mu)}{n_{max}^{-\mu}}, \\
&= |\mathcal{A}|n_{max}^{\mu}\zeta(\mu), \\
&\leq |\mathcal{A}||\mathcal{X}|^{\mu}\zeta(\mu),
\end{aligned}
$$

thus satisfying our assumption. On the other hand, if we let the duration distribution be geometric, $z(n) = \lambda(1 - \lambda)^{n-1}$, we have

$$
\begin{aligned}
\frac{1}{\rho(\omega)} &\leq \frac{|\mathcal{A}|}{z(n_{max})}, \\
&= \frac{|\mathcal{A}|}{\lambda(1 - \lambda)^{n-1}}, \\
&= \frac{1}{\lambda}|\mathcal{A}|\left(\frac{1}{1 - \lambda}\right)^{n_{max}-1}. \\
&\leq \frac{1}{\lambda}|\mathcal{A}|\left(\frac{1}{1 - \lambda}\right)^{|\mathcal{X}|-1}.
\end{aligned}
$$

As $1/(1 - \lambda) > 1$ and $n_{max}$ is only bounded by the number of states, this results in an upper bound that is exponential in the number of states and therefore does not satisfy the assumptions.

## B  DOMAIN SPECIFICATIONS

**DeepSea (Osband, Aslanides, and Cassirer, 2018)**  Parameterized by problem size $N$, this environment can be viewed as the lower triangle of an $N \times N$ gridworld with two actions: "down-left" and "down-right" which move diagonally down either left or right. There is a single goal state in the far bottom-right corner, which can only be reached through a single action-sequence. The goal reward is $1.0$, and there is a per-step reward of $-0.01/N$. Finally, all episodes end after exactly $N$ steps, once the agent reaches the bottom. Therefore, the maximum possible undiscounted return is $0.99$. An example with $N = 20$ is shown in Figure 6a. Average first-passage times are shown for a problem size of $N = 20$ in Figure 3a, and unlike other plots are logarithmically scaled, $\log(\mathbb{E}\,[\text{fpt}] + 1)$ with contour levels in the range $[0, 16]$.

In this work we use the deterministic variant of DeepSea; however, the standard stochastic version *randomizes* the action effects at every state. That is, "down-left" may correspond to action index $0$ in one state and $1$ in another, and these assignments are performed randomly for each training run (consistently across episodes). We briefly mention this variant in our conclusions as an example in

which our proposed method should be expected to perform poorly. Indeed, in Figure 14 we show that such an adversarial modification reduces $\epsilon z$-greedy's performance back to that of $\epsilon$-greedy.

For experiments, we used Q-learning with a tabular function approximator, learning rate $\alpha = 1.0$, and $\epsilon = 1.0/(N+1)$ for problem size $N$. Experiment results are averages over 30 random seeds.

**GridWorld** Shown in Figure 6b, this is an open single-room gridworld with four actions ("up", "down", "left", and "right"), and a single non-zero reward at the goal state. The initial state is in the top center of the grid (offset from the wall by one row), and the goal state is diagonally across from it at the other end of the room. Notice that if the goal were in the same row or column, as well as if it were placed directly next to a wall, this could be argued to advantage an action-repeat exploration method. Instead, the goal location was chosen to be harder for $\epsilon z$-greedy to find (offset from wall, far from and not in same row/column as start state).

For experiments, we used Q-learning with a tabular function approximator, learning rate $\alpha = 0.1$, $\epsilon = 0.1$, and maximum episode length 1000. Experiment results are averages over 30 random seeds.

Figure 1 shows average first-passage times on a similar gridworld, but with a fixed greedy policy which takes the "down" action deterministically.

**MountainCar (Sutton and Barto, 2018)** This environment models an under-powered car stuck in the valley between two hills. The agent must build momentum in order to reach the top of one hill and obtain the goal reward. In this version of the domain all rewards are zero except for the goal, which yields reward of $1.0$. There are two continuous state variables, corresponding to the agent location, $x$, and velocity, $\dot{x}$.

The dense-reward version of this environment can be solved reliably in less than a dozen episodes using linear function approximation on top of a low-order Fourier basis (Konidaris, Osentoski, and Thomas, 2011).

In our experiments using the sparse-reward variant of the environment, we used SARSA($\lambda$) with a linear function approximation on top of an order 5 Fourier basis. We used learning rate $\alpha = 0.005$, $\epsilon = 0.05$, $\gamma = 0.99$, and $\lambda = 0.9$. The maximum episode length was set to 5000. Experiment results are averages over 30 random seeds. A near-optimal policy, given this discount and $\epsilon$, but without confounds due to function-approximation, should reach approximately $0.29$ episodic discounted return.

**CartPole (Barto, Sutton, and Anderson, 1983)** We use the "swingup_sparse" variant as implemented in Tassa et al. (2018). In this sparse reward version of the environment, the agent receives zero reward unless $|x| < 0.25$ and $\cos(\theta) > 0.995$, for the cart location $x$ and pole angle $\theta$. All episodes run for 1000 steps, and observations are 5-dimensional continuous observation.

For experiments, we used SARSA($\lambda$) with a linear function approximation on top of an order 7 Fourier basis. We used learning rate $\alpha = 0.0005$, $\epsilon = 0.01$, $\gamma = 0.99$, and $\lambda = 0.7$. The maximum episode length was 1000. Weights were initialized randomly from a mean-zero normal distribution with variance 0.001. Experiment results are averages over 30 random seeds.

**Atari-57** (Bellemare et al., 2013), is a benchmark suite of 57 Atari 2600 games in the Arcade Learning Environment (ALE). Observations are $210 \times 160$ color images (following Mnih et al. (2015), in many agents these are down-scaled to $84 \times 84$ and converted to grayscale). For the primary results in this work we use the original ALE version of Atari 2600 games, which does not include subsequently added games (beyond the 57) or features such as "sticky actions". For results with sticky actions enabled consult Appendix E.

Many existing results on Atari-57 report performance of the best agent throughout training, or simply the maximum evaluation performance attained during training. We do not report this metric in the main text because it does not reflect the true learning progress of agents and tends to reflect an over estimate. However, for comparison purposes, "best" performance is included later in the Appendix. In the next section, alongside other agent details, we will give hyper-parameters used in the Atari-57 experiments. An example frame from the game PRIVATE EYE is shown in Figure 6e.

## C Agent and Algorithm Details

Except for ablation experiments on the duration distribution, all $\epsilon z$-greedy experiments use a duration distribution $z(n) \propto n^{-\mu}$ with $\mu = 2.0$. These durations were capped at $n \leq 10000$ for all experiments except for the Rainbow-based agents which were limited to $n \leq 100$, but in this case no other values were attempted.

### Pseudo-code

---
**Algorithm 1** $\epsilon z$-Greedy exploration policy

---
1: **function** EZGREEDY($Q$, $\epsilon$, $z$)
2:     $n \leftarrow 0$
3:     $\omega \leftarrow -1$
4:     **while** True **do**
5:         Observe state $x$
6:         **if** $n == 0$ **then**
7:             **if** random() $\leq \epsilon$ **then**
8:                 Sample duration: $n \sim z$
9:                 Sample action: $\omega \sim U(\mathcal{A})$
10:                 Assign action: $a \leftarrow \omega$
11:             **else**
12:                 Greedy action: $a \leftarrow \arg\max_a Q(x, a)$
13:         **else**
14:             Assign action: $a \leftarrow \omega$
15:             $n \leftarrow n - 1$
16:         Take action $a$

---

### Network Architecture.

**Rainbow-based agents** use an identical network architecture as the original Rainbow agent (Hessel et al., 2018). In particular, these include the use of NoisyNets (Fortunato et al., 2017), with the exception of Rainbow-CTS, which uses a simple dueling value network like the "no noisy-nets" ablation in Hessel et al. (2018). A preliminary experiment showed this setting with Rainbow-CTS performed slightly better than when NoisyNets were included.

**R2D2-based agents** use a slightly enlarged variant of the network used in the original R2D2 (Kapturowski et al., 2019), namely a 4-layer convolutional neural network with layers of 32, 64, 128 and 128 feature planes, with kernel sizes of 7, 5, 5 and 3, and strides of 4, 2, 2 and 1, respectively. These are followed by a fully connected layer with 512 units, an LSTM with another 512 hidden units, which finally feeds a dueling architecture of size 512 (Wang et al., 2015). Unlike the original R2D2, Atari frames are passed to this network without frame-stacking, and at their original resolution of $210 \times 160$ and in full RGB. Like the original R2D2, the LSTM receives the reward and one-hot action vector from the previous time step as inputs.

### Hyper-parameters and Implementation Notes

Unless stated otherwise, hyper-parameters for our Rainbow-based agents follow the original implementation in Hessel et al. (2018), see Table 2. An exception is the Rainbow-CTS agent, which uses a regular dueling value network instead of the NoisyNets variant, and also makes use of an $\epsilon$-greedy policy (whereas the baseline Rainbow relies on its NoisyNets value head for exploration). The $\epsilon$ parameter follows a linear decay schedule 1.0 to 0.01 over the course of the first 4M frames, remaining constant after that. Evaluation happens with an even lower value of $\epsilon = 0.001$. The same $\epsilon$-schedule is used in Rainbow+$\epsilon$-greedy and Rainbow+$\epsilon z$-greedy, *on top* of Rainbow's regular NoisyNets-based policy.

The CTS-based intrinsic reward implementation follows Bellemare et al. (2016), with the scale of intrinsic rewards set to a lower value of 0.0005. This agent was informally tuned for better

performance on hard-exploration games: Instead of the "mixed Monte-Carlo return" update rule from Bellemare et al. (2016), Rainbow-CTS uses an $n$-step Q-learning rule with $n = 5$ (while the baseline Rainbow uses $n = 3$), and differently from the baseline does not use a target network.

All of our R2D2-based agents are based on a slightly tuned variant of the published R2D2 agent (Kapturowski et al., 2019) with hyper-parameters unchanged, unless stated otherwise - see Table 3. Instead of an $n$-step Q-learning update rule, our R2D2 uses expected SARSA($\lambda$) with $\lambda = 0.7$ (Van Seijen et al., 2009). It also uses a somewhat shorter target network update period of $400$ update steps and the higher learning rate of $2 \times 10^{-4}$. For faster experimental turnaround, we also use a slightly larger number of actors (320 instead of 256). This tuning was performed on the vanilla R2D2 in order to match published results.

The RND agent is a modification of our baseline R2D2 with the addition of the intrinsic reward generated by the error signal of the RND network from Burda et al. (2018). The additional networks ("predictor" and "target" in the terminology of Burda et al. (2018)) are small convolutional neural networks of the same sizing as the one used in Mnih et al. (2015), followed by a single linear layer with output size 128. The predictor is trained on the same replay batches as the main agent network, using the Adam optimizer with learning rate 0.0005. The intrinsic reward derived from its loss is normalized by dividing by its variance, utilizing running estimates of its empirical mean and variance. Note, the RND agent includes the use of $\epsilon$-greedy exploration.

The Bootstrapped R2D2 agent closely follows the details of Osband et al. (2016). The R2D2 network is extended to have $k = 8$ action-value function heads which share a common convolutional and LSTM network, but with distinct fully-connected layers on top (each with the same dimensions as in R2D2). During training, each actor samples a head uniformly at random, and follows that action-value function's $\epsilon$-greedy policy for an entire episode. Each step, a mask is sampled, and added to replay, with probability $p = 0.5$ indicating which heads will be trained on that step of experience. During evaluation, we compute the average of each head's $\epsilon$-greedy policy to form an ensemble policy that is followed.

| **Rainbow** (baseline) | |
|---|---|
| Replay buffer size | $10^6$ observations |
| Priority exponent | 0.5 |
| Importance sampling exponent | annealed from 0.4 to 1.0 |
| | (over the course of 200M frames) |
| Multi-step returns $n$ | 3 |
| Discount $\gamma$ | 0.99 |
| Minibatch size | 32 |
| Optimiser | Adam |
| Optimiser settings | learning rate $= 6.25 \times 10^{-5}$, $\varepsilon = 1.5 \times 10^{-4}$ |
| Target network update interval | 2000 updates (32K environment frames) |
| $\epsilon$ (training) | 0.0 (i.e. no $\epsilon$-greedy used) |
| $\epsilon$ (evaluation) | 0.0 (i.e. no $\epsilon$-greedy used) |
| | |
| **Rainbow+$\epsilon$/$\epsilon z$-greedy, Rainbow+CTS** | |
| $\epsilon$ (training) | **linear decay from 1.0 to 0.001** |
| | **(over the course of 4M frames)** |
| $\epsilon$ (evaluation) | **0.001** |
| | |
| **Rainbow+CTS only** | |
| Multi-step returns $n$ | **5** |
| Intrinsic reward scale ($\beta$ in Bellemare et al. (2016)) | **0.0005** |
| Target network update interval | **1 (i.e., no target network used)** |

Table 2: Hyper-parameters values used in Rainbow-based agents (deviations from Hessel et al. (2018) highlighted in boldface).

| | |
|---|---|
| Number of actors | **320** |
| Actor parameter update interval | 400 environment steps |
| Sequence length | 80 (**+ prefix of 20 for burn-in**) |
| Replay buffer size | $4 \times 10^6$ observations ($10^5$ part-overlapping sequences) |
| Priority exponent | 0.9 |
| Importance sampling exponent | 0.6 |
| Learning rule | **Expected SARSA($\lambda$), $\lambda = 0.7$** |
| Discount $\gamma$ | 0.997 |
| Minibatch size | 64 |
| Optimiser | Adam |
| Optimiser settings | **learning rate $= 2 \times 10^{-4}$, $\varepsilon = 10^{-3}$** |
| Target network update interval | **400 updates** |

Table 3: Hyper-parameters values used in R2D2-based agents (deviations from Kapturowski et al. (2019) highlighted in boldface).

| Environment | # Trials | # Steps | Max Episode Length | Contour Scale | Discretization |
|---|---|---|---|---|---|
| DeepSea | 30 | $500000 \times N$ | N | Log | None |
| GridWorld | 100 | 5000 | 5000 | Linear | None |
| MountainCar | 50 | 5000 | 5000 | Linear | 12 |
| CartPole | 100 | 5000 | 5000 | Linear | 20 |

Table 4: Settings for experiments used to generate average first-visit visualizations found in main text.

## D   EXPERIMENT DETAILS

**First-visit visualizations**   These results (e.g. see Figure 1) are intended to illustrate the differences in state-visitation patterns between $\epsilon$-greedy and $\epsilon z$-greedy. These are generated with some fixed $\epsilon$, often $\epsilon = 1.0$ for pure-exploration independent of the greedy policy, and are computed using Monte-Carlo rollouts with each state receiving an integer indicating the first step at which that state was visited on a given trial. States that are never seen in a trial receive the maximum step count, and we then average these over many trials. For continuous-state problems we discretize the state-space and count any state within a small region for the purposes of visitation. We give these precise values in Table 4.

**Atari experiments**   The experimental setup for the Rainbow-based and R2D2-based agents each match those used in their respective baseline works. In particular, Rainbow-based agents perform a mini-batch gradient update every 4 steps and every 1M environment frames learning is frozen and the agent is evaluated for 500K environment frames. In the R2D2-based agents, acting, learning, and evaluating all occur simultaneously and in parallel, as in the baseline R2D2.

In the Atari-57 experiments, all results for Rainbow agents are averaged over 5 random seeds, while results for R2D2-based agents are averages over 3 random seeds.

Atari-57 is most often used with a built-in number (4) of action-repeats for every primitive action taken (Mnih et al., 2015). We did not change this environment parameter, which means that an exploratory action-repeat policy of length $n$ will, in the actual game, produce $4 \times n$ low-level actions. Similarly, DQN-based agents typically use frame-stacking, while agents such as R2D2 which use an RNN do not. The robustness of our results across these different algorithms suggests that $\epsilon z$-greedy is not greatly impacted by the presence or absence of frame-stacking.

**Atari metrics**   The human-normalized score is defined as

$$score = \frac{\text{agent} - \text{random}}{\text{human} - \text{random}},$$

where agent, random, and human are the per-game scores for the agent, a random policy, and a human player respectively. The *human-gap* is defined as the average, over games, performance difference

Figure 7: **Stochastic** Gridworld experiments. **(a)** We show averaged training performance (over 100 episodes) with respect to the noise scale. **(b)** Example learning curves from these experiments showing the effect of stochasticity on both agents.

between human-level over all games,

$$human\_gap = 1.0 - \mathbb{E} \min(1.0, score).$$

**Computational Resources** Small-scale experiments were written in Python and run on commodity hardware using a CPU. Rainbow-based agents were implemented in Python using JAX, with each configuration (game, algorithm, hyper-parameter setting) run on a single V100 GPU. Such experiments generally required less than a week of wall-clock time. R2D2-based agents were implemented in Python using Tensorflow, with each configuration run on a single V100 GPU and a number of actors (specified above) each run on a single CPU. These agents were trained with a distributed training regime, described in the R2D2 paper (Kapturowski et al., 2019), and required approximately 3 days to complete.

# E EXPERIMENTAL RESULTS: STOCHASTICITY

In this section we present a series of experiments focused on the effects of stochasticity and non-determinism in the environment on the performance of $\epsilon z$-Greedy.

## SMALL-SCALE ENVIRONMENTS

We consider stochastic versions of two of our small-scale environments: GridWorld, and MountainCar. In the Gridworld domain the "noise scale" is the probability of the action transitioning to a random neighboring state. In MountainCar, where actions apply a force in $-1, 0, 1$ to the car, we add mean zero Gaussian noise to this force with variance given by the specified noise scale. Finally, note that these forces are clipped to be within the original range of $[-1, 1]$.

Figure 7a shows the discounted return, averaged over the 100 episode training runs, as a function of the noise scale. Figure 7b gives example learning curves for the agents ($\epsilon$-Greedy and $\epsilon z$-Greedy) for four levels of noise. We first note that for near-deterministic settings we replicate the original findings in the main text, but that the performance of the $\epsilon$-Greedy agent actually improves for small amounts of transition noise, while both agents degrade performance as this noise becomes larger. We interpret this to suggest that the $\epsilon$-Greedy agent is initially benefiting from the increased exploration, whereas $\epsilon z$-Greedy was already exploring more and begins to degrade slightly sooner.

Figure 8 similarly shows the two agent's performance on MountainCar as we increase the scale of the transition noise. Here we see both agents generally suffer reduced performance as the level of noise increases and is maximal for both in the deterministic case, unlike in Gridworld.

## ATARI-57 WITH STICKY-ACTIONS

The Arcade Learning Environment (Bellemare et al., 2013, ALE), supports a form of non-determinism called *sticky actions* where with some probability $\zeta$ (typically set to $0.25$) the agent's action is ignored and instead the previously executed action is repeated (Machado et al., 2018a). This is not exactly equivalent to simpler transition noise, such as we used in the small-scale experiments above, because there is now a non-Markovian dependence on the previous action executed in the environment. Two additional details are important to keep in mind. First, the agent does not observe the action that was executed, and instead only observes the action that it intended to take. Second, the sticky-action effect applies at every step of the low-level ALE interaction. That is, it is standard practice to use

Figure 8: **Stochastic** MountainCar experiments. **(a)** We show averaged training performance (over 1000 episodes) with respect to the noise scale. **(b)** Example learning curves from these experiments showing the effect of stochasticity on both agents.

| Algorithm (@200M) | Median | Mean | Human-gap |
|---|---|---|---|
| Rainbow | 1.81 | 13.47 | 0.172 |
| Rainbow+$\epsilon$-Greedy | 2.10 | 14.13 | 0.152 |
| Rainbow+$\epsilon z$-Greedy | **2.17** | **14.53** | **0.141** |

Table 5: **Sticky-action** Atari-57 final performance summaries for Rainbow-based agents after 200M environment frames.

an action-repeat (usually $4$) in Atari, such that each action selected by the agent is repeated some number of times in the low-level interaction. Thus, sticky actions apply at every step of this low-level interaction.

Because of the non-Markovian effects and the relatively fast decay in probability of action-repeats, sticky actions do not provide similar exploration benefits as seen in $\epsilon z$-Greedy. Unlike in exploration using action-repeats the agent is unable to learn about the actions that were actually executed, making the underlying learning problem more challenging. In some environments, where precise control is needed for high performance, sticky actions tend to significantly degrade performance of agents.

We note that sticky actions are a modification usually applied to the problem, not the agent, and usually make the problem harder. The reason for this is that sticky actions increase the level of stochasticity of the environment, but do so with a non-Markovian dependency on the previously executed action. Given the superficial similarity with action-repeats, one might ask whether using sticky actions might benefit exploration on hard exploration games similar to $\epsilon z$-Greedy. In order to answer this question we compare our Rainbow-based $\epsilon$-Greedy and $\epsilon z$-Greedy agents on the Atari-57 benchmark with sticky actions enabled ($\zeta = 0.25$).

Figure 9 gives the median, mean, and human-gap values for the human-normalized scores. We observe that much of the gap in performance in mean and median have disappeared, and that even for human-gap the performance benefits of $\epsilon z$-Greedy over $\epsilon$-Greedy have been partially reduced. Nonetheless, we continue to see significant improvements in performance on the same set of harder exploration games as in the non-sticky-action results (see Figure 10). Finally, we give the numeric values for the final performance of these agents in Table 5.

## F  EXPERIMENTAL RESULTS: LIMITATIONS

To further study the limitations of $\epsilon z$-Greedy, and motivate work on learned options for exploration, we consider the effect on performance of adding "obstacles" and "traps" in the Gridworld domain. For our purposes, an obstacle is an interior wall in the gridworld, such that if the agent attempts to occupy the location of the obstacle the result is a no-op action with no state transition. On the other hand, a "trap" results in the immediate end of an episode, with zero reward, if the agent attempts to occupy the same location as the trap. Figure 11 shows a set of example gridworlds with obstacles and traps generated at varying target densities. We generate the environments by filling a $20 \times 20$ gridworld with either obstacles or traps (not both), where each state has some probability of being so filled (given by the target density). We then identify the largest connected component of open cells and select the goal and start states randomly without replacement from the states in this component. This ensures that all gridworlds can be solved.

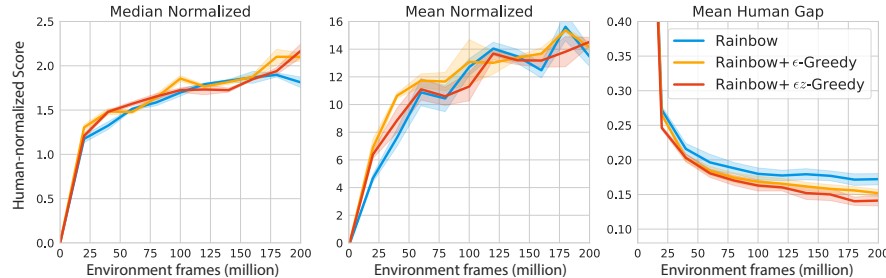

Figure 9: **Sticky-action** Atari-57 summary curves Rainbow-based agents.

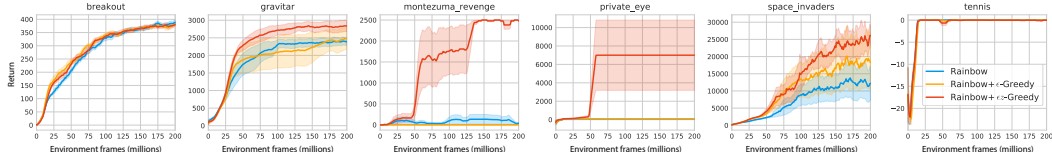

Figure 10: **Sticky-action** Atari-57 results for selected games showing Rainbow-based agents. Curves are averages over 3 seeds.

Note that increasing the density of obstacles and traps has two opposite effects: on one hand it reduces the overall state space, thus making the problem easier, on the other hand it also makes exploration more difficult, since obstacles result in a sparser transition graph and traps result in many absorbing states. Based on our observations, the latter effect tends to be stronger than the former.

Our experiments focus on studying the performance of $\epsilon$-Greedy and $\epsilon z$-Greedy as we scale the density of the obstacles or traps. Figure 12 & 13 show our results in these two sets of experiments. In both cases we observe that the gap in performance between $\epsilon z$-Greedy and $\epsilon$-Greedy decreases with the density of the problematic states. Interestingly, we see that $\epsilon$-Greedy is not impacted as seriously. We believe this is partly to be expected because $\epsilon$-Greedy is exploring more locally and more densely around the start state, making navigating around obstacles somewhat easier. Note that we have increased the number of episodes from 100 (used in the main text) to 1000 in order to increase the likelihood of both agents solving a given problem. Additionally, unlike in the main text, every trial of these experiments is performed on a randomly generated gridworld with randomly selected start and goal locations; although we do ensure that each agent is trained on the same environment, the environment itself is different for each seed. This is reflected in the larger variances in performance indicated by the shaded regions in the figures.

## G  FURTHER EXPERIMENTAL RESULTS

In this section, we include several additional experimental results that do not fit into the main text but may be helpful to the reader. In the conclusions we highlight a limitation of $\epsilon z$-greedy which occurs when the effects of actions differ significantly between states. In Figure 14 we present results for such an adversarial setting in the DeepSea environment, where the action effects are randomly permuted for every state. We observe, as expected, that in this setting $\epsilon z$-greedy no longer provides more efficient exploration than $\epsilon$-greedy.

In Figure 15 we compare with RMax (Brafman and Tennenholtz, 2002) on the Gridworld domain. We consider two values for the threshold for a state-action being marked as *known*: $N = 1$, effectively encoding an assumption that the domain is deterministic, and $N = 10$ which is a more generally reasonable value. We observe that, if tuned aggressively, RMax can significantly outperform $\epsilon z$-Greedy, as should be expected. However, we note that unlike RMax $\epsilon z$-Greedy does not assume access to a tabular representation and can scale to large-scale problems.

Next, in Figures 16 & 17 we report the per-game percentage of relative improvement, using human-normalized scores, over an $\epsilon$-greedy baseline for $\epsilon z$-greedy in both the Rainbow and R2D2 agents.

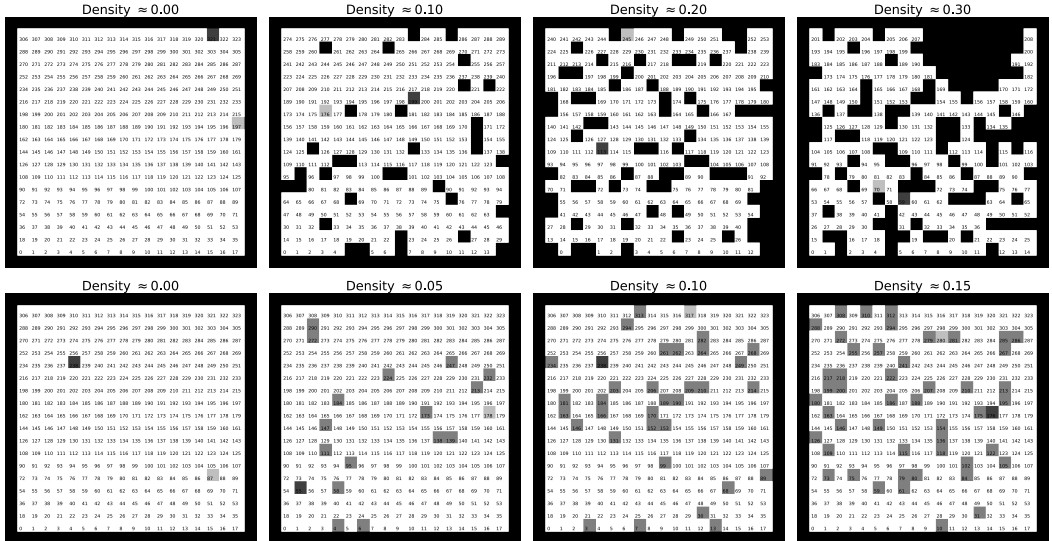

Figure 11: Example Gridworlds with varying density of **(top)** obstacles and **(bottom)** traps. The shades of grey represent the type of each cell: white cells are open states, light grey cells are the start state, grey cells are traps, dark grey cells are goal states, and black cells are obstacles. These environments are randomly generated to a target density of obstacle / trap, while ensuring there exists a path between the start and goal states.

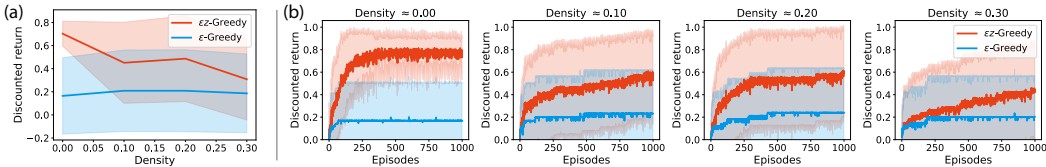

Figure 12: Gridworld with obstacles at varying density. **(a)** We show averaged training performance (over 1000 episodes) with respect to the obstacle density. **(b)** Example learning curves from these experiments showing the effect on both agents.

In both cases we give the corresponding results for improvement of CTS, for Rainbow, and RND, for R2D2, over the same baselines. Additionally, we give these results for both the final agent performance and the performance averaged over training. The percent relative improvement of a *score* over a *baseline* is computed as

$$100 \times \frac{score - baseline}{baseline}.$$

Note that we limit the maximum vertical axis in such a way that improvement in some games is cut off. This is because for some games these relative improvements are so large that it becomes difficult to see the values for other games on the same scale.

In the main text we give summary learning curves on Atari-57 for Rainbow- and R2D2-based agents in terms of median human-normalized score and human-gap. In Figure 18 we show these as well as the mean human-normalized score learning curves. In Figures 19 & 20 we give full, per-game results for Rainbow- and R2D2-based agents respectively. These results offer additional context on those we reported in the main text, demonstrating more concretely the nature of the performance trade-offs being made by each algorithm.

Finally, in Table 1 we give mean and median human-normalized scores and the human-gap on Atari-57 for the final trained agents. However, this is a slightly different evaluation method than is often used (Mnih et al., 2015; Hessel et al., 2018), in which only the best performance for each game, over training, is considered. For purposes of comparison we include these results in Table 6.

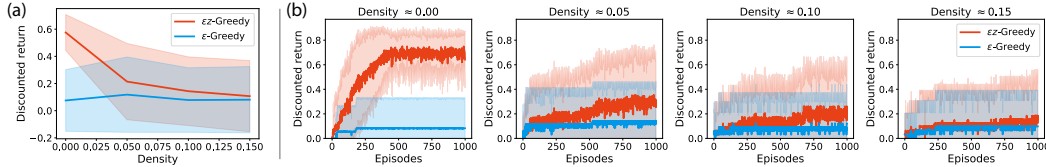

Figure 13: Gridworld with traps at varying density. **(a)** We show averaged training performance (over 1000 episodes) with respect to the trap density. **(b)** Example learning curves from these experiments showing the effect on both agents.

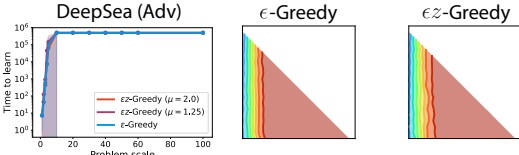

Figure 14: Adversarial modification to DeepSea environment causes $\epsilon z$-greedy to perform no better than $\epsilon$-greedy.

| Algorithm (@30B) | Median | Mean | Human-gap | Median (best) | Mean (best) |
|---|---|---|---|---|---|
| R2D2 | 14.31 | 39.55 | 0.102 | 19.36 | 46.98 |
| R2D2+RND | 9.40 | **42.17** | 0.151 | 14.34 | **48.02** |
| R2D2+Bootstrap | 15.75 | 37.69 | 0.096 | 19.35 | 43.87 |
| R2D2+$\epsilon z$-greedy | **16.64** | 40.16 | **0.077** | **22.63** | 45.33 |
| Algorithm (@200M) | | | | | |
| Rainbow | 2.09 | 8.82 | 0.139 | 2.20 | 12.24 |
| Rainbow+$\epsilon$-Greedy | 2.26 | 9.17 | 0.144 | 2.56 | 12.23 |
| Rainbow+CTS | 1.72 | 6.77 | 0.157 | 2.09 | 7.62 |
| Rainbow+$\epsilon z$-Greedy | **2.35** | **9.34** | **0.130** | **2.74** | **12.28** |

Table 6: Atari-57 final performance summaries. R2D2 results are after 30B environment frames, and Rainbow results are after 200M environment frames. We also include median and mean human-normalized scores obtained by using *best* (instead of *final*) evaluation scores for each training run, to allow comparison with past publications which often used this metric (e.g. Hessel et al. (2018)).

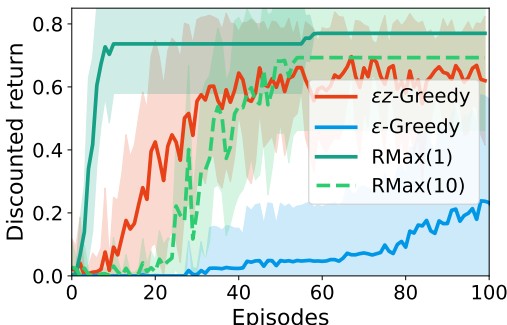

Figure 15: Experiment in the Gridworld domain comparing Rmax, with visitation thresholds 1 and 10 with $\epsilon$-Greedy and $\epsilon z$-Greedy.

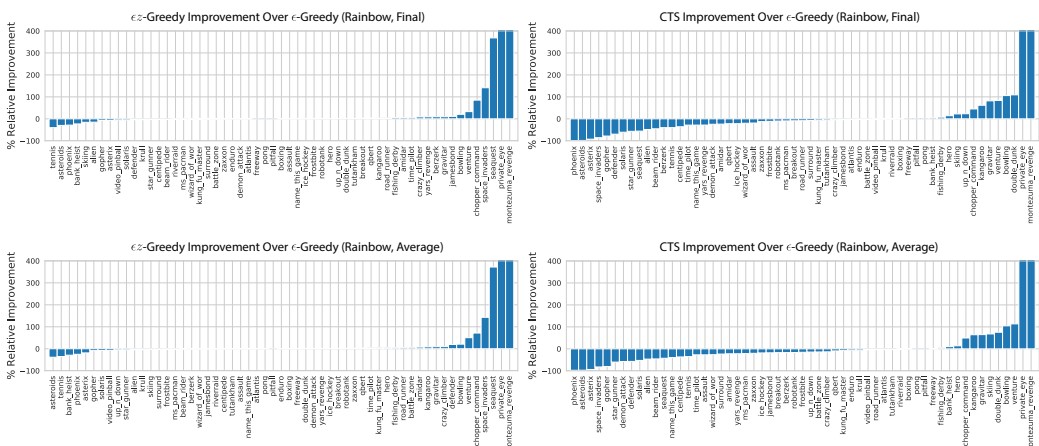

Figure 16: Percent relative improvement of exploration methods ($\epsilon z$-Greedy and CTS) over $\epsilon$-Greedy for the Rainbow-based agents on Atari-57 per-game. We report this for both final performance (top) and average over training (bottom).

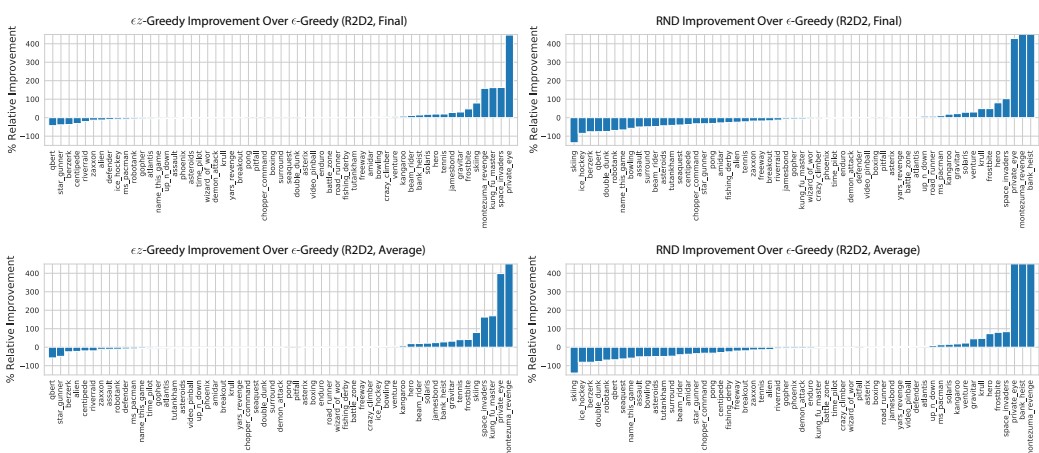

Figure 17: Percent relative improvement of exploration methods ($\epsilon z$-Greedy and RND) over $\epsilon$-Greedy for the R2D2-based agents on Atari-57 per-game. We report this for both final performance (top) and average over training (bottom).

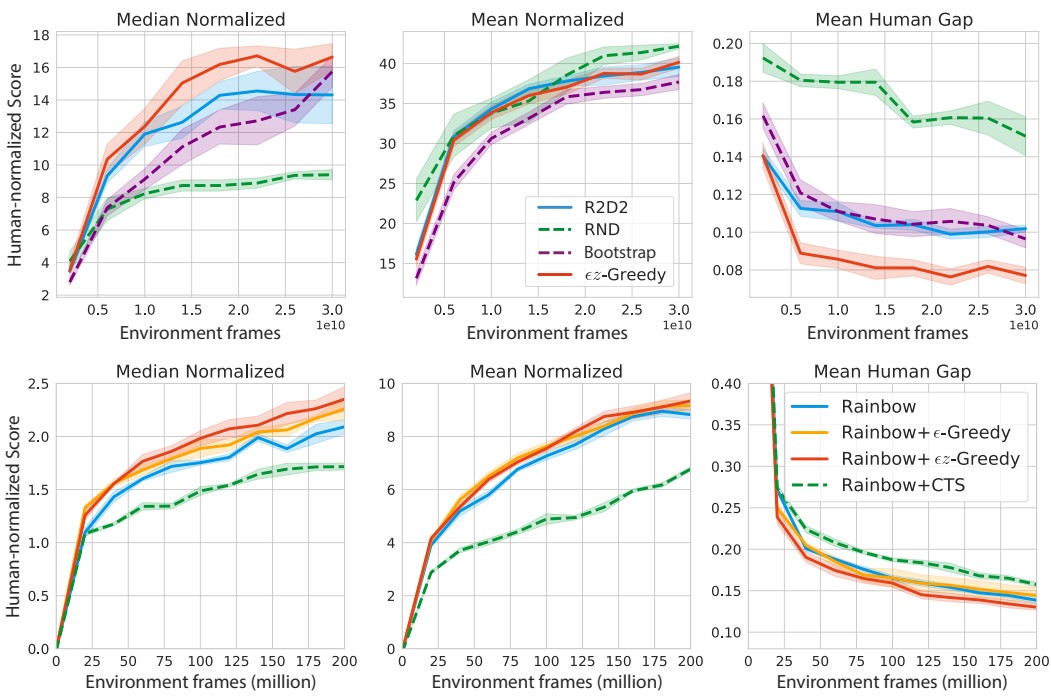

Figure 18: Atari-57 summary curves for R2D2-based methods (top) and Rainbow-based methods (bottom).

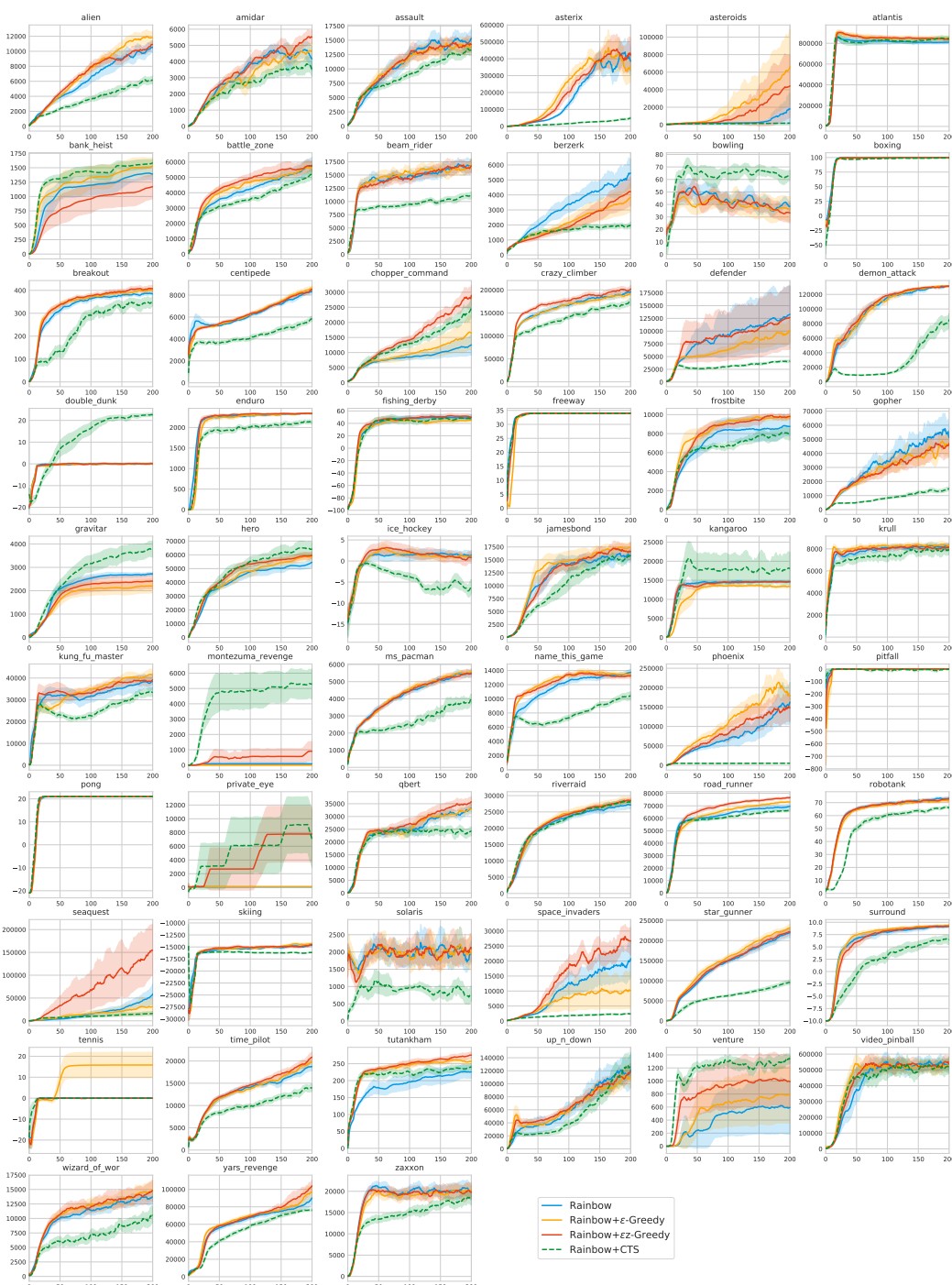

Figure 19: Per-game Atari-57 results for Rainbow-based methods.

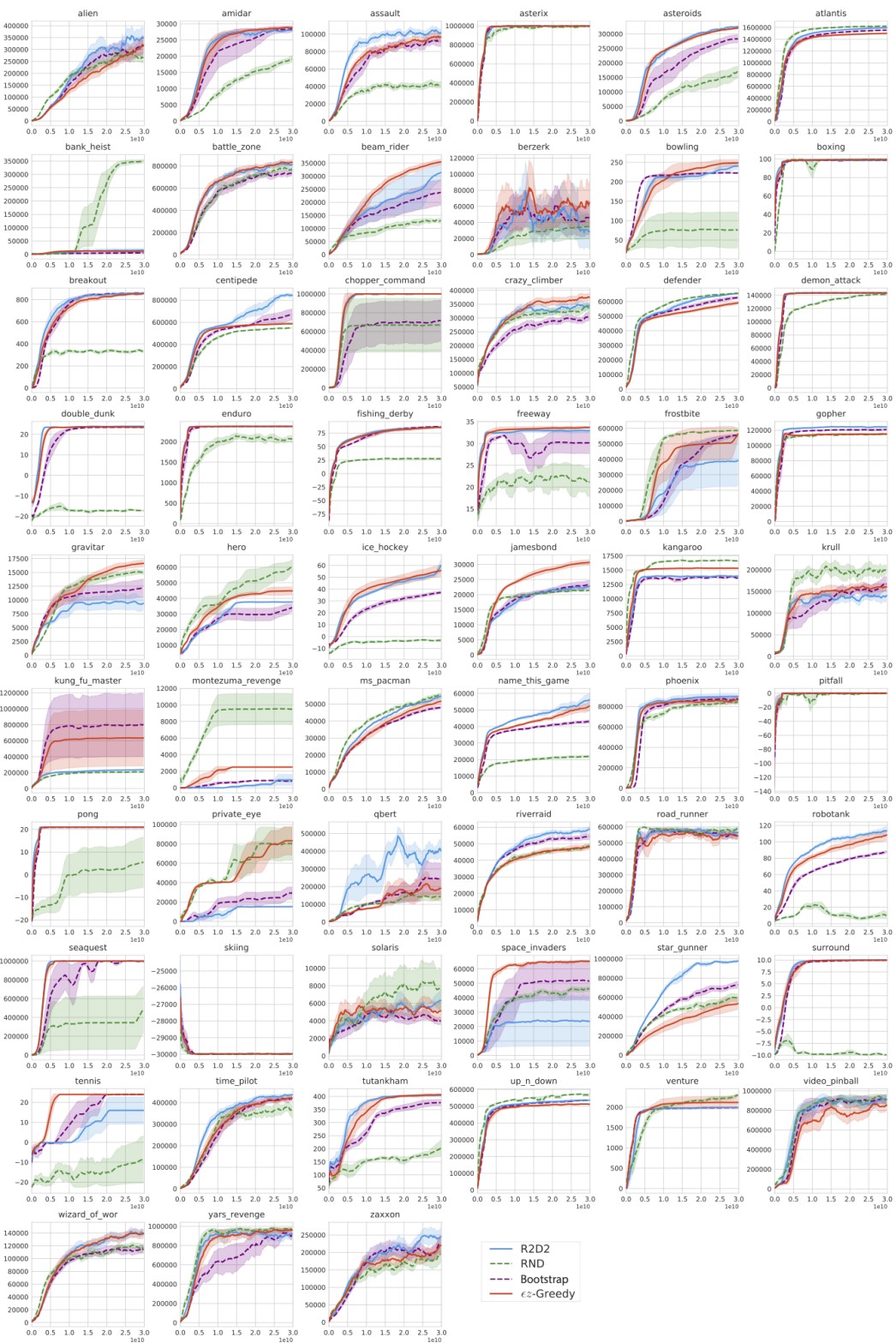

Figure 20: Per-game Atari-57 results for R2D2-based methods.

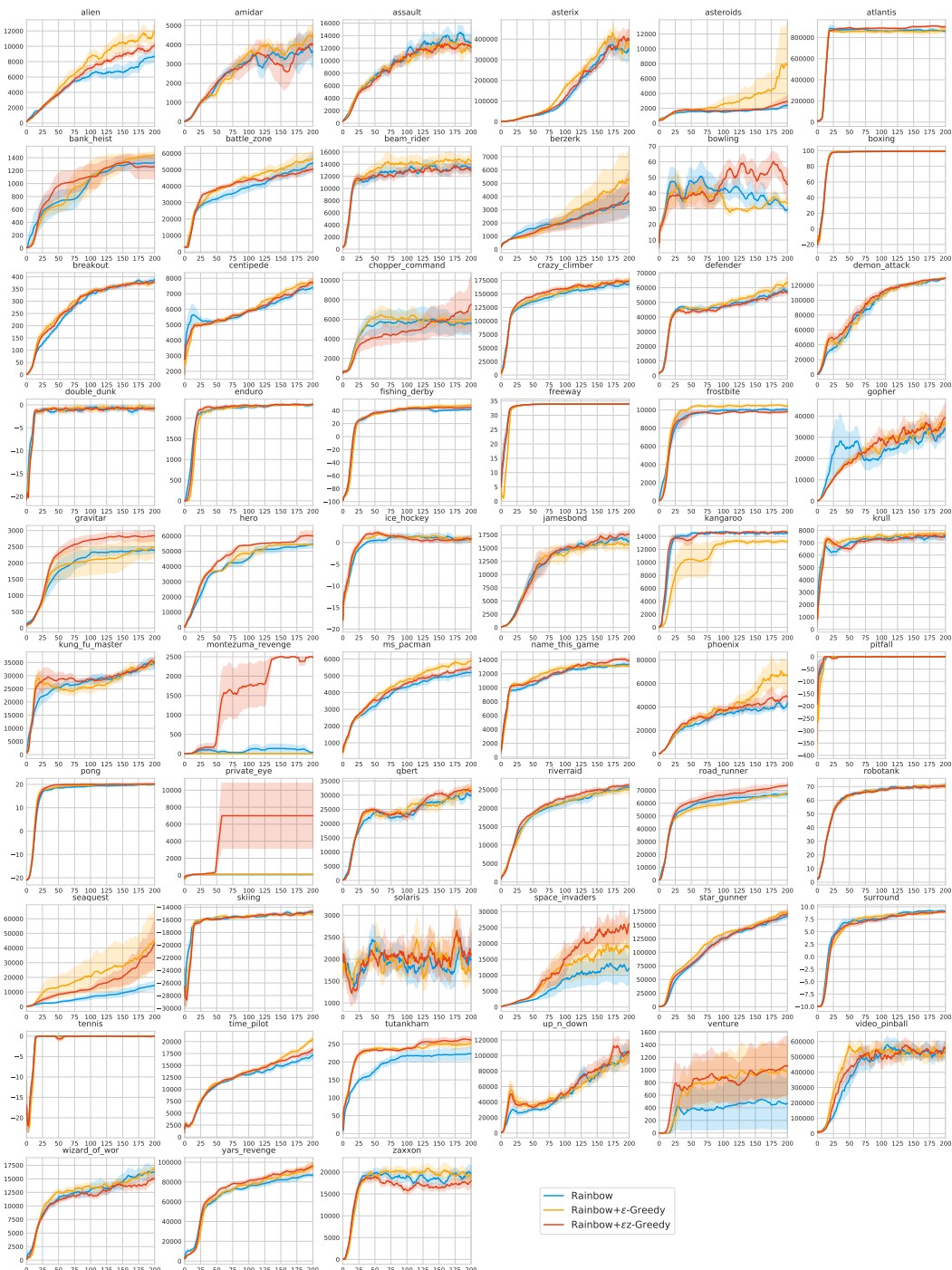

Figure 21: Per-game **Sticky-action** Atari-57 results for Rainbow-based methods.

