# OpenReview forum: "Temporally-Extended ε-Greedy Exploration"
_ICLR.cc/2021/Conference — ICLR 2021 Poster_

### Official Review · AnonReviewer3 · 2020-10-27
**A simple algorithm.**

**Rating:** 6
**Confidence:** 4

**Review:**

This paper proposes an easy-to-implement algorithm for the efficient exploration, which is a temporally-extended version of \eps-greedy. Instead of uniformly selecting primitive actions with probability \eps for exploration, the algorithm explores using options. In theory, it has been shown that if the option set is well-designed with a sublinear expected reaching time, the algorithm achieves a polynomial sample complexity. Empirically, the authors tested a simple instantiation, ez-greedy, in multiple environments and claimed that ez-greedy improves exploration and performance in sparse-reward environments with minimal loss in performance on easier, dense-reward environments.

I appreciate this work for its motivation and the algorithm is simple-to-implement and not computationally expensive, which points to an interesting direction for future study.

But there are several concerns I have:
1. The conditions in Thm. 1, i.e. a sublinear upper bound on visiting time and 1/p(w), is not straightforward for me to realize. How should one construct such an option set if no prior knowledge is given? Do the option-learning methods in [Jinnai et al. 2019, 2020] and [Machado et al. 2017 2018] satisfy these conditions? Does ez-greedy satisfy these conditions? If not, what approximately are the upper bounds for these heuristics? There should be more discussion about this.
2. In the tabular RL, it would be more complete if the authors can compare with UCB-based exploration strategies as well, e.g. the UCB-Q as in http://papers.nips.cc/paper/7735-is-q-learning-provably-efficient.
3. As mentioned by the authors, the performance of ez-greedy depends on whether the effects of actions differ significantly across states. There should be more adversarial cases to show the possible outcomes of ez-greedy compared with other exploration strategies.
4. As mentioned by the authors, action-repeats are not new in deep RL. The novelty of this work lies in using it for exploration with sampled duration. However, the selection of zeta-distribution for duration sampling is only partially empirically demonstrated as well as the choice of \mu. It would be better if the authors can support it with a theoretical justification or some quantitive analysis in terms of e.g., how the value of \mu affects the final performance.

I am open to adjusting the score if the rebuttal can address my concerns.

---

> ### Author Response · Authors · 2020-11-18
> **Response to Reviewer #3**
>
> Thank you for your review and careful consideration of the work. We will attempt to directly address each of your four concerns below.
>
> 1. The conditions on Theorem 1 may be satisfied, or not, for some combination of a set of options, sampling distribution, and MDP. That is, and perhaps this is your point, knowing if these conditions will hold or not does require some (perhaps significant) prior knowledge about the MDP under consideration. For example, ez-Greedy (which defines the options and sampling distribution) on the standard Chain MDP (in which one of the actions progresses and the other returns to the start state) does satisfy these conditions. But, conversely, if we replace the duration distribution of ez-Greedy with an exponential this is no longer the case because it introduces an exponential dependence on the number of states in the chain. We will add a discussion of this example with the math worked out in Appendix A in a future revision. Additionally, we will attempt to establish sufficient conditions under which the Covering Options of Jinnai et al. (2019) would satisfy Assumption 1 and report back once we have a concrete answer.
>
> 2. This is a fair concern as our small-scale experiments do not compare with any other baselines than epsilon-greedy. We will attempt to add results for a UCB-based Q-Learning algorithm on the two tabular domains (DeepSea and Gridworld) to Appendix E in a future revision. However, notice that such an algorithm is not immediately applicable to the non-tabular environments, and that the pseudo-counts-based exploration (Rainbow+CTS) was an attempt, by Bellemare et al. (2016), to extend such methods to neural network based RL.
>
> 3. To address this concern we are extending our small-scale problems to include additional Gridworlds with features that make them more 'adversarial' to the ez-Greedy method. There are two additional Gridworlds: (1) which places obstacles throughout the world which severely limits the utility of action-repeats, and (2) which places random traps throughout the world which penalize over-exploration of known-bad state-actions. We will include these in a revision as soon as possible and post a message notifying you of the update.
>
> In particular, we will report the performance of ez-Greedy as these disadvantage properties are increased, showing how performance degrades as the assumptions are violated more dramatically.
>
> 4. We hope that our additional result and discussion around your first point will help to motivate the use of the zeta distribution. Additionally, note that Figure 6b (Appendix) shows the performance of the ez-Greedy R2D2-based agent as the mu hyperparameter is varied.

---

> > ### Comment · AnonReviewer3 · 2020-11-19
> > **Look forward to new experiment results**
> >
> > I want to thank the authors for their efforts to address my concerns. I hope these revisions can be posed to the submission before Nov. 24th and then I can adjust my score accordingly.

---

> > > ### Author Response · Authors · 2020-11-24
> > > **Updated revision**
> > >
> > > Thank you, we have updated the revision, and hope to upload a final revision before the deadline. We have added the Chain MDP example and discussion in Appendix A of the updated version. We are currently working on determining sufficient conditions for covering options to satisfy Assumption 1. We hope to include this before the deadline.
> > >
> > > We have also added a new set of experiments in Appendix F (Experimental Results: Limitations), where we investigate the effect on the gridworld results of adding obstacles and traps at varying density levels to the environment. As we explain in detail in the section, these are done by using procedurally generated gridworlds with obstacles (which block the agent) or traps (which end the episode) at varying densities, while also randomizing start and goal locations. We see that ez-greedy is indeed negatively affected by these modifications, although not as catastrophically as we might have feared. In general, we see ez-greedy performance degrading with increased density until eventually matching that of epsilon-greedy.
> > >
> > > We are still working on providing UCB-Q results as requested by the reviewer. We have verified that an RMax agent performs as expected, which is generally better than either epsilon- or ez-greedy. And we are now trying to obtain similar results for the requested method. The superior performance of RMax in this setting is expected, because (1) these are small, tabular, environments where exact counts are possible and (2) this method eventually stops exploring and will thus show better final performance if the competing methods are not adjusted accordingly. We expect the results of UCB-Q to exhibit a similar trend. We highlight that, unlike the proposed algorithm, these methods do not scale well to large or infinite state spaces.

---

> > > > ### Comment · AnonReviewer3 · 2020-11-24
> > > > **New Score**
> > > >
> > > > I would like to raise my score to 6 for the following reasons:
> > > > 1. this algorithm is simple-to-implement and practical for deep RL;
> > > > 2. it brings more attention to exploration with simple options;
> > > > 3. the paper is well-written and clear;
> > > > 4. the authors added extra experiments to address my second and third concerns;
> > > > 5. it provides more thoughts on how we evaluate exploration strategies: overfitting to hard tasks or improving over easier tasks in general.
> > > >
> > > > I did not give a higher score since:
> > > > 1. more theoretical understanding is needed, e.g., the reason for choosing zeta-distribution;
> > > > 2. the action-repeat strategy is not new;
> > > > 3. the action-repeat strategy can be domain-specific.

---

> > > > > ### Author Response · Authors · 2020-11-24
> > > > > **Thank you**
> > > > >
> > > > > Thank you for updating your score, we really do appreciate your active engagement. We have updated the revision with some additional information and results based upon your comments and questions.
> > > > >
> > > > > In thinking more carefully about your question regarding Theorem 1, we realized that we made a notational mistake in the statement of the theorem. Where you see “O” it should be “\Theta”. This means that the bound is any polynomial in |X| and |A| rather than a polynomial order 1 (linear). We hope this alleviates your concern, since building such a set of options seems more realizable. The worked example on the Chain MDP in which ez-greedy satisfies the assumption may be illustrative. We also argue that, as one adds more covering options, they will eventually satisfy Assumption 1.
> > > > >
> > > > > Finally, we note that Theorem 1 is only one possible way of showing that temporally-extended epsilon-greedy is feasible. The results in the Jinnai et al. paper actually provide another line of argument, as their notion of expected cover time applies directly to temporally-extended epsilon-greedy, and in their case is paired with the option learning algorithm to minimize this quantity.
> > > > >
> > > > > Given the time left to the deadline we do not think we will be able to get UCB-Q working successfully, but have included our experiments with RMax to provide some additional context in relative performance in the tabular Gridworld domain. We compare epsilon-greedy and ez-greedy with Rmax with the ‘known’ threshold set to 1 (assuming the environment is deterministic) and 10, and observe that ez-greedy compares reasonably well with the latter. These results can be found in Figure 14 (in Appendix G).

---

### Official Review · AnonReviewer4 · 2020-10-27
**Insufficient impact**

**Rating:** 5
**Confidence:** 4

**Review:**

The paper presents an extension of \eps-greedy strategy in order to increase the coverage of exploration in RL problems. The main idea is to take an exploratory option instead of a single action e.g., by repeating an action for n steps, where the duration of repeat is sampled from some distribution. The authors demonstrate that given certain conditions, the algorithm will converge in polynomial time for Q-learning method.

Overall, the paper provides a simple yet effective exploration technique for RL methods. However, there are some unclear points regrading the impact of the approach on the filed, I thus vote for a weak reject.

Pros:
- A simple and universal approach to promote full coverage in reinforcement learning
- Scalable and easy to implement
- Interesting inspiration from animal foraging behavior
- Demonstrates effective performance in conducted experiments


Comments:

- The key concern about the paper is whether gaining the full coverage in exploration is beneficial at any cost. In my view, RL favors smarter exploration over the total coverage. For instance , the exemplary scenario in figure 1 shows how the proposed approach increases the coverage of the state space.  But why we would need to search the whole space if we are on the right path to the goal. Unnecessary coverage will lead to higher regret and delayed convergence that is the result of naive exploration.

- Another problem with \eps-greedy is that it explores forever. Hence, it would be an improvement to stop/reduce exploration at some point rather than intensify it. Assume in the steps close to the goal, the worst action has the same probability to be chosen as the second-best action and it repeats for n times which moves the agent farther from the goal. I would suggest to add evaluation in terms of regret and/or convergence time in such scenarios.

- The limitations both enumerated in the paper and the dependency of the approach on some strong conditions on the options seem to be more than the benefits of the proposed method. Besides, it is not general in terms of class of applications; e.g., in complex tasks we can not simply repeat an action in different states.

- How does the approach behave in very simple settings, e.g., bandits? I would recommend to add such analysis for online exploration into the paper.

Minor:
Table 1 is not referenced in the text

---

> ### Author Response · Authors · 2020-11-18
> **Response to Reviewer #4**
>
> Thank you for your review and for bringing up some interesting points for discussion. The main points you raise focus on the nature of any form of temporally-extended epsilon-greedy, with the final two perhaps mostly applied to ez-Greedy specifically.
>
> We now (in the current revision) reference Table 1 in the main text. Thank you for catching this.
>
> The cost of full coverage: We argue that full coverage (in the limit) is important, as without this property we are assuming some MDPs will simply never be solved. You are correct that this property comes at a cost, and thus there is a trade-off between generality and efficiency to be made. Indeed, one might argue that our work is moving (gradually) from generality towards efficiency, when compared with epsilon-greedy. Regarding the example in Figure 1, if we know that the agent is following the optimal policy then there is no point in further exploration. However, if we don't know this, then the current path could be sub-optimal or going in the completely wrong direction. Without fully exploring the state space the agent would never know whether or not the current policy is optimal/sub-optimal.
>
> Epsilon-greedy, and ez-Greedy, explore forever: When used in practice the epsilon value is often annealed over time, but the essential point is completely valid. Both epsilon-greedy, and temporally-extended epsilon-greedy, continue to explore with some probability indefinitely. This is something we would argue is out-of-scope for this work, despite being an important direction for future work.
>
> Bandits: It is unclear how any form of temporally-extended exploration would benefit exploration in bandits, due to the lack of a temporal dependency on previous actions. Perhaps the concept could be stretched to cover this setting, but it feels like a poor fit.
>
> "in complex tasks we can not simply repeat an action in different states.": This was our initial thought as well, but the 'complexity' of tasks in which such a simple set of policies are beneficial is larger than we expected (as evidenced by our empirical results). That said, we agree with the spirit of your point here: ez-Greedy will not be effective on all MDPs. We discuss three ways in which ez-Greedy exploration can be detrimental in our section “Generality and Limitations,” but would welcome insights into what other aspects of a complex domain would make ez-Greedy unsuitable. With additional work on option-learning, we believe the use of temporally-extended epsilon-greedy on such environments, where the learned options capture, instead of assume, the structure of the problem, will be quite effective.

---

### Official Review · AnonReviewer2 · 2020-10-27

**Rating:** 8
**Confidence:** 5

**Review:**

##########################################################################
Summary:

This paper proposes a simple yet general approach for exploration in discrete-action problems. The proposed approach, called ez-greedy, combines randomly selected options with the well-adopted e-greedy exploration policy to achieve temporally-extended e-greedy exploration. The paper overviews the publicized exploration methods from the perspective of their inductive biases, and clearly states where the inductive bias of ez-greedy would be better suited over e-greedy. The paper reports results in tabular, linear, and deep RL settings, on numerous domains ranging from classic toy problems to Atari-57. The results are interesting, and the analysis aligns and supports nicely the narrative of the paper.

##########################################################################
Reasons for Score:

Overall, I vote for accepting this paper. The idea is simple (a generalization of e-greedy) and the discussions nicely illustrate the main properties of an ideal generally-applicable exploration method. The experiments clearly show where ez-greedy exploration would be useful. Also, they show that the inductive bias of ez-greedy does not hurt much the performance in simpler dense-reward domains while more specialized algorithms suffer significantly.

##########################################################################
Pros:

See "Reasons for Score" above.

##########################################################################
Cons:

1) The results in Atari are based on a deterministic version of Atari (i.e. not using "sticky actions"). Also, in DeepSea the deterministic version of the task is used. Ideally, I would've liked to see empirical results in stochastic domains as well. More importantly, I'm not sure why only deterministic domains are used?

2) The literature on action-repeats are discussed briefly. But it's hard to know how the former related works were different in their formulation and use of action-repeats. Also, could you clarify how sticky-actions are positioned w.r.t. ez-greedy (beyond that the purpose behind sticky-actions was to induce stochasticity in the environment as opposed to being used explicitly for exploration)? For instance, do sticky-actions actually improve learning performance in the same domains were ez-greedy improves performance?

3) The rainbow + e-greedy vs. Rainbow + ez-greedy Median and Mean plots do not show significant findings. I think a bar-plot should be added to show per-game relative human-normalized improvements for these versions. The same should be done for R2D2 (e-greedy) vs. R2D2 + ez-greedy as well. I think what this could reveal is symmetric bars over the 57-Atari games (i.e. number of games in which ez-greedy outperforms and underperforms e-greedy are the same). Also, the extent of improvements on average is the same as shown in the Mean plot of Figure 8.
To clarify, I don't see an issue with this outcome (i.e. if the bars are symmetric; meaning overall there are as many games in Atari-57 that would benefit from ez-greedy over e-greedy as there are games in which the opposite is the case). This does not go against the narrative of the paper which makes it clear that they each have an inductive bias that suits some tasks over others. But I think this should be made super clear in the results section, through such bar plots. For the same reason, I think the Mean plots should also be brought to the main text and shown next to the Median curves.

4) Why only 5 random seeds in DeepSea? I suggest showing results for 30 randoms seeds like in the other toy problems.

##########################################################################
Questions during the rebuttal period:

Please address and clarify the "Cons" above.

##########################################################################
Minor comments:

- It would be useful to replace "Rainbow" with "Rainbow (NoisyNet)" in Figure 3 so as to emphasize the difference between "Rainbow" and "Rainbow + e-greedy". Similarly, for "R2D2" it'd make it easier for the reader if the Figures show "R2D2 (e-greedy)".

- Table 1: "Algorithm (@200M)": M doesn't need to be italicized (to be consistent with "Algorithm (@30B)").

- It'd make it easier if "(100%)" is added to the y-axis of Median/Mean plots.

---

> ### Author Response · Authors · 2020-11-18
> **Response to Reviewer #2**
>
> Thank you for your detailed review, positive assessment and constructive feedback. We have incorporated your other minor comments in the current revision, thank you again, and below attempt to address your primary comments.
>
> 1. Deterministic domains: This is a fair concern and we are in the process of running a subset of our Atari experiments, and the entirety of our small-scale experiments, on stochastic versions of the domains - we don't expect ez-Greedy to be particularly affected by this and are positive the additional evaluation will confirm this intuition. We will post an additional message when we have updated the revision with these results.
>
> 2. Regarding sticky-actions, once we have completed the above stochasticity experiments we will include additional discussion regarding the relation between sticky-actions and the action-repeats used by ez-Greedy. Briefly, sticky-actions do not, in general, improve performance. When exploring with action-repeats (and options in general) the agent observes the actions it takes and is thus able to learn about all the states and actions in an exploratory trajectory. With sticky-actions the agent only observes that action which they *intended* to take, not the one actually executed by the environment. Finally, sticky-actions have fairly short duration due to having an exponential decay in probability. A related concept which can improve performance is changing the base action-repeat number (generally set to 4 in Atari). This value can absolutely be tuned in a domain-dependent manner to improve performance, although the conventional value still appears to be the best fixed choice. However, even here there is a notable difference between using action-repeats (also, options) for exploration, and learning the values of those action-repeat (option) policies and using this for credit assignment and planning. Game-dependent tuning of action-repeats would confound all three effects (exploration, credit assignment, planning), while we can say with more certainty that ez-Greedy is only directly affecting exploration.
>
> 3. Yes, the error bars for Rainbow+epsilon-Greedy and Rainbow+ez-Greedy partially overlap. We now include in Appendix E the requested bar-plots showing % improvement per-game. Please note, however, that the objective here is to improve on the challenging domains (requiring more exploration) without degrading performance overall. To make this point a bit clearer we have also included the same bar-plots for Rainbow-CTS and R2D2-RND. The results do indeed show that ez-Greedy suffers a small degradation in performance on a small number of games, while offering fairly large improvements on a larger number of games (this pattern holds in both agent settings but differ in the specifics). Meanwhile, we observe that for CTS and RND the number and magnitude of the games for which performance is degraded is much larger, helping to provide context for the summary plots shown in the main text.
>
> 4. The number of seeds was chosen only to match that of published results, we are running additional seeds and will update the DeepSea results to be over 30 seeds in a future revision.
>
> Regarding the "Rainbow (NoisyNet)" label, we are open to this, but want to check with the reviewer on whether they still believe this to be the clearest way of labeling the methods. All the Rainbow-based agents we considered except for Rainbow+CTS include NoisyNets. We remark on this choice in the paper, as it was found that NoisyNets had a small negative effect on the Rainbow+CTS agent. Thus, Rainbow (e-greedy) is actually Rainbow (NoisyNets + e-greedy). Do you still think your proposed naming is the best choice? Another option to improve clarity would be to drop the "Rainbow" and "R2D2" prefix entirely for these plots and specify only the exploration in each case. This seems cleanest, and we would appreciate the reviewer's thoughts on this.

---

> > ### Comment · AnonReviewer2 · 2020-11-21
> > **Thanks for the clarifications and added plots - looking forward to your new results!**
> >
> > Thank you for the clarifications regarding sticky-actions and the added bar plots, which I found very useful.
> >
> > * Minor comment: In the new bar plots you may be clipping the top performance bars at ~400%.
> >
> > * Regarding relabeling to clarify the plots: I see your point, so I think the way you have it currently is fine.
> >
> > I look forward to seeing your results in stochastic problems.

---

> > > ### Author Response · Authors · 2020-11-24
> > > **Updated results**
> > >
> > > The latest revision includes (in Appendix E) new experiments and discussion around the effects of stochasticity. This includes a series of experiments on two of the small-scale domains (Gridworld and Mountaincar) where we systematically vary the amount of transition noise in the environment and compare epsilon-greedy and ez-greedy. These results are quite interesting and we hope the reviewer will find them informative. The performance characteristics in these two environments show interesting similarities and differences as the amount of noise is scaled up.
> > >
> > > We have also now included experimental results (also in Appendix E) on the sticky-action version of Atari, where we compare only Rainbow-based epsilon-greedy and ez-greedy (due to time constraints). We note that, although the benefits of ez-greedy in terms of the summary statistics is slightly reduced with sticky actions (Figure 9), we still observe significant performance improvement on the harder exploration games (Figure 10). In this section we have also included a more detailed discussion of sticky-actions and their relation to ez-greedy action-repeats. Please note that not all of our seeds (3 per agent) have completed in all games, and thus we cut the summary statistics plot at 175 million frames. We will update the figures with the full 200 million frames before the deadline.
> > >
> > > DeepSea results are now over 30 seeds for all experimental results presented. The difference between the original and updated plots is minimal.
> > >
> > > Regarding the bar plots being clipped at 400%. You are correct, we clipped the plots here because for both ez-greedy and the other exploration methods the percent improvement for a few games is so large as to make all other bars much too small to see when plotted together. We could report the numeric values in the legend if this is preferable, though one can also observe the difference in performance in these games in the per-game plots shown in Figure 18 of Appendix G.

---

### Official Review · AnonReviewer1 · 2020-11-01
**An important problem which is weakly supported by a concrete model**

**Rating:** 5
**Confidence:** 4

**Review:**

This paper presents a generalized overview of temporally extended e-greedy exploration. Basic principle of temporally extended e-greedy exploration is to apply the e-greedy exploration policy for an extended period of time. Specifically, authors use a heavy-tailed zeta distribution.

Strong points
- This paper analyze theoretical properties of temporally extended e-greedy exploration in Theorem 1.
- ez-Greedy policy outperforms e-Greedy policy in some experiments environments qualitative and quantitatively.

Weak points
- The theoretical analysis is too general under too strong assumptions. Thus, the presented results are not unexpectedly novel.
- The presented methods based on the zeta distribution are not concrete enough.
- Algorithm is not clearly specified. Thus, it is hard to evaluate the algorithmic contributions of this paper.

Although this paper presents a general analysis on temporally extended e-greedy exploration, the presented ideas are too general. Thus, it is very hard to verify the technical contributions (in terms of models and algorithms) of this paper.

---

> ### Author Response · Authors · 2020-11-18
> **Response to Reviewer #1**
>
> Thank you for taking the time to read and review our work. We focus on addressing the weak points you have brought up.
>
> Theoretical analysis: We certainly want theoretical results that are as general as possible under the weakest possible assumptions. Although the impact of our theoretical results will be somewhat limited by their assumptions, they still serve as an important contribution by illustrating how the components of temporally-extended epsilon-greedy influence sample complexity. We believe this adds significantly to the paper because it helps to provide intuition about when option-based exploration will be beneficial.
>
> We were unsure regarding the second two weak points you raised. The concrete method, ez-Greedy, is extremely simple and specified in pseudo-code in the appendix. Would you be willing to clarify so we can better address them?
>
> The duration is sampled from a zeta distribution, which has implementations in many statistical packages (e.g. numpy.random.zipf), but can also be easily approximated with access to the Riemann zeta function (for normalization). In this case a duration n is sampled with probability n^(-mu) / Z(mu), where Z is the Riemann zeta function. We may have misunderstood the reviewer's concern, so please consider replying to clarify.
>
> We would suggest that this concrete algorithm (ez-Greedy) is less an algorithmic contribution and more a simple instantiation of the more general temporally-extended epsilon-greedy approach discussed in the first part of the paper.

---

### Official Review · AnonReviewer5 · 2020-11-06

**Rating:** 8
**Confidence:** 4

**Review:**

**Summary:**

This paper offers a critique of current exploration techniques as being overly complex and engineered to only work on specific tasks. As an alternative, the paper proposes temporally extended $ \epsilon$-greedy exploration which maintains the simplicity and generality of $ \epsilon$-greedy while offering better . More specifically, the proposed algorithm simply repeats the randomly chosen action for a random number of steps (where the number of steps comes from a specific distribution), this is a specific instantiation of the more general algorithm presented in the paper where any set of semi-markov options can be used.

--------------------------------------------------------------------

**Strengths:**

1. Clarity. This paper is very well-written and clear, making it enjoyable to read. It sets up the shortcomings of prior methods and offers a simple solution. I also especially appreciated the clear discussion of the limitations of the proposed method.

2. Strong critique of prior methods to provide motivation. It is an important observation that while many exploration methods are developed in the theory and deep RL communities, they are often inferior in practice to simple strategies like $ \epsilon$-greedy. While this is not a novel contribution, this paper really drives home the point by providing a slightly smarter variant of dithering that competes favoriable with much more complicated algorithms. This is an especially important contribution of this paper since it makes the point to the RL community that simple exploration strategies may be more effective in practice, but there is still room to innovate while maintaining simplicity and generality.

3. Strong empirical results. The experiments clearly show an improvement over $ \epsilon$-greedy in small benchmark problems. Then, they demonstrate how $ \epsilon z$-greedy even improves over more complocated exploration strategies for deep RL algorithms applies to atari relative to more complicated exploration algorithms like RND (at least in the "average" case, but not on "hard exploration" games like Montezuma's revenge).



--------------------------------------------------------------------

**Weaknesses:**

1. The theory could be tightened. The paper would be stronger if the theorem were stated more formally (defining polynomial sample complexity) and the proof provided the specific results being used from the cited papers (maybe as lemmas in the appendix). At a more substantive level, it is not clear how exhaustive the list of desired properties of an exploration algorithm is. The paper lists three desiderata for an exploration strategy: (1) that it is simple, (2) that it is stationary, and (3) that it promotes full coverage of the state-action space. Each of these goals makes sense, but the paper does not provide any framework to explain why these are a necessary or sufficient set of properties to yield the desired behavior. Moreover, it is not clear what the tension or tradeoffs are between the properties. A more clear discussion of these issues or formal framework could go a long way toward clarifying the landscape of exploration algorithms.

--------------------------------------------------------------------

**Recommendation:**

I reccomend accepting this paper and gave it a score of 8. I think the paper provides a clear argument for simple and general exploration strategies and that $ \epsilon z$-greedy seems to be an algorithm that achieves these goals. Moreover, I think that the paper makes an important point to the community working on exploration algorithms that the complicated algorithms being developed can often be beaten by simple strategies when considering a broad range of problems.

--------------------------------------------------------------------

**Additional feedback:**

- One reference that I think should be included when discussing learning temporally extended representations of actions is [1].

- Typo: line 2 of the last paragraph on page 1 should be "such a compromise".
- The discussion of the choice of distribution over durations was somewhat abrupt. This is an interesting part of the algorithm and it would be nice if it was fleshed out a bit more.



[1] Whitney, W., Agarwal, R., Cho, K., & Gupta, A. (2019). Dynamics-aware Embeddings. *arXiv preprint arXiv:1908.09357*.

---

> ### Author Response · Authors · 2020-11-18
> **Response to Reviewer #5**
>
> Thank you, we appreciate your positive feedback and suggestions where the paper can be improved. The revised paper fixes the typo and adds the suggested citation, which is highly relevant for this work considering the way they factor the embedding of state and action-sequences, and the property of near-uniformity in state transitions from sampled action embeddings. Furthermore, there may be some hope that learning *this* form of option, though not framed as such, could be done more efficiently than in some of the other discussed approaches due to the embedding being state independent. Given the additional space available for revision we will work on extending the discussion around duration distribution for a future revision.
>
> Your point about the limitations in the theoretical results are well taken. We expect to add an additional result and discussion that may be relevant to this point (in response to another reviewer) to Appendix A in a future revision and will call your attention to it once available. We are also working on strengthening the theoretical work along the lines of your suggestion, but this is not yet reflected in the current revision. Once we have incorporated this into a revision we will post a message to that effect.

---

### Decision · Program_Chairs · 2021-01-07
**Final Decision**

**Decision:**

Accept (Poster)

**Comment:**

This paper proposes a simple generalization to epsilon-greedy exploration that induces temporally extended probes and can leverage options.  The idea and analysis are trivial.  Computational results demonstrate when this sort of exploration is helpful.  The paper is well written and the authors offer a fair assessment of when these ideas do or do not address challenging exploration tasks.  A range of computational results support and offer insight into the concepts.